# Identification, Quantification, and Characterization of HIV-1 Reservoirs in the Human Brain

**DOI:** 10.3390/cells11152379

**Published:** 2022-08-02

**Authors:** Maribel Donoso, Daniela D’Amico, Silvana Valdebenito, Cristian A. Hernandez, Brendan Prideaux, Eliseo A. Eugenin

**Affiliations:** Department of Neuroscience, Cell Biology, and Anatomy, University of Texas Medical Branch (UTMB), Research Building 17, Fifth Floor, 105 11th Street, Galveston, TX 77555, USA; mdonoso@utmb.edu (M.D.); dadamico@utmb.edu (D.D.); sivaldeb@utmb.edu (S.V.); craherna@utmb.edu (C.A.H.); brpridea@utmb.edu (B.P.)

**Keywords:** NeuroHIV, anti-retroviral, cure, survival, HIV-1, dementia, reservoirs

## Abstract

The major barrier to cure HIV infection is the early generation and extended survival of HIV reservoirs in the circulation and tissues. Currently, the techniques used to detect and quantify HIV reservoirs are mostly based on blood-based assays; however, it has become evident that viral reservoirs remain in tissues. Our study describes a novel multi-component imaging method (HIV DNA, mRNA, and viral proteins in the same assay) to identify, quantify, and characterize viral reservoirs in tissues and blood products obtained from HIV-infected individuals even when systemic replication is undetectable. In the human brains of HIV-infected individuals under ART, we identified that microglia/macrophages and a small population of astrocytes are the main cells with integrated HIV DNA. Only half of the cells with integrated HIV DNA expressed viral mRNA, and one-third expressed viral proteins. Surprisingly, we identified residual HIV-p24, gp120, nef, vpr, and tat protein expression and accumulation in uninfected cells around HIV-infected cells suggesting local synthesis, secretion, and bystander uptake. In conclusion, our data show that ART reduces the size of the brain’s HIV reservoirs; however, local/chronic viral protein secretion still occurs, indicating that the brain is still a major anatomical target to cure HIV infection.

## 1. Introduction

HIV/AIDS represents a major global public health concern, with an estimated 38 million infected individuals worldwide and 1.2 million in the US. Despite the significant success of antiretroviral treatments in controlling systemic HIV replication, these therapies are not a cure. The main obstacle to cure HIV-infected individuals is the existence of latently infected cells in multiple tissues that persist for extended periods despite long-term antiretroviral therapy (ART) [1,2,3,4].

HIV has been shown to persist within different cell types and compartments of the body, including blood, lymph nodes, platelets/megakaryocytes/bone marrow, urethra, and other immune-privileged sites, such as the brain [5,6,7,8,9,10,11]. However, the size, identity, and localization of these viral reservoirs, as well as the mechanisms of long-term survival, are still under active investigation [12,13,14,15,16]. Currently, most viral reservoir detection methodologies focus on a single viral component, such as viral DNA, mRNA, episomal DNA, viral proteins, or associated viral replication. The identification of these viral components depends on artificial in vitro cloning and reactivation methodologies resulting in significant limitations in accuracy, sensitivity, cost, testing times, and the requirement for a large blood volume from patients [17,18,19,20,21,22]. Currently, limited methodologies exist to detect or quantify residual viral replication and protein expression within most tissues. The objective of our study is to provide a robust methodology for simultaneous detection and identification of these reservoirs using tissues derived from patients during the pre-ART era, as well as from ART-adherent patients.

In the periphery (mostly blood products), several studies have identified that the primary HIV reservoir resides in naïve and memory CD4^+^ T lymphocytes; however, several groups propose that circulating reservoirs may be a poor reflection of the actual/real viral reservoirs present in tissues [12,13,15,18,23,24,25,26,27,28]. In the human brain, laser-captured material and subsequent determination of integrated HIV DNA using Alu-PCR demonstrate that brain microglia/macrophages and a small population of astrocytes harbor HIV DNA [29,30,31,32]. New technical developments, such as deep sequencing and more sensitive PCR techniques further confirmed the presence of integrated HIV DNA in several tissues, including the brain [1,12,13,14,15,26,33]. Overall, the consensus is that microglia/macrophages can act as viral brain reservoirs. It is still debated whether astrocytes are also viral reservoirs due to their low infectivity and replication [34,35,36,37,38,39,40,41]. Only recently has it been demonstrated that HIV-infected astrocytes allow HIV egress into the periphery, demonstrating that although these reservoirs are low in numbers and highly compartmentalized, they can still repopulate the entire body with the virus [33]. Thus, eliminating all viral reservoirs is required to achieve an HIV cure, including those present in the brain.

Here, we contribute to this knowledge by developing a multi-component imaging-based methodology to identify, quantify, and characterize rare and low replicating HIV reservoirs in patient samples (blood and tissues) with unprecedented accuracy and reliability compared to DNAscope, RNAscope, viral protein staining, and several PCR-related techniques or systemic measures of replication. Our findings using large brain areas showed that microglia/macrophages and astrocytes are the brain’s viral reservoirs in the pre-ART, early stages of ART, and long-term ART era. The long-lasting ART reduced the abundance of HIV-infected cells but did not prevent the residual production of HIV-mRNA and viral proteins in all the cases analyzed. Additionally, we identified that glial viral reservoirs are more stable than in microglia/macrophages, suggesting that long-term ART has differential effects on myeloid and glial reservoirs. In addition, residual viral protein production resulted in the secretion and bystander uptake by uninfected cells in close contact with HIV-infected cells. Thus, residual viral replication and active viral protein secretion into neighboring cells could explain the chronic inflammation observed in the HIV-infected population even in the current ART era. However, long-term ART reduced the expression of some viral proteins, but not all, in a cell type-specific manner, suggesting that each cell type control residual viral replication differently or could be less susceptible to ART. Both mechanisms need further investigation to identify the common and distinct mechanisms. Our data will certainly lead to breakthroughs in the HIV field and guide efforts to eliminate CNS compromise in the HIV-infected population and obtain a cure for HIV. 

## 2. Materials and Methods

**A protocol for the simultaneous detection of integrated HIV DNA, viral mRNA, and HIV proteins in brain tissue obtained from HIV-infected individuals.** The evolution of this protocol has been described in several publications depending on the tissue, blood, or experimental condition analyzed [33,42,43,44]. Overall, the advantage of this protocol is the simultaneous detection of integrated HIV DNA (HIV DNA), viral mRNA (HIV-mRNA), viral proteins, and several cellular markers in the same assay. We describe all the details below using various tissues obtained from HIV-infected individuals (see Table 1 for patient data).

The detection system is based on the following premises: for the detection of integrated HIV DNA, first, the automatic or manual detection of cells that are positive for HIV DNA, and second, the HIV DNA probe needs to colocalize with DAPI, and Alu-repeat staining at least a 0.8 Pearson’s correlation coefficient as described using specific regions of interest (ROI) [45]. These two conditions are essential to consider a signal as integrated HIV DNA, or the signal is considered negative or unspecific. For the detection of HIV-mRNA, first, low colocalization with DAPI or Alu repeats was required (0.2 Pearson’s correlation coefficient or below), and second, the presence of the mRNA was only present in cells with HIV DNA signal. These points are based on the fact that in all our studies, no cells negative for HIV DNA were positive for HIV-mRNA. To detect viral proteins, we included both instead cell types in the analyses, HIV DNA positive and negative to quantify bystander protein uptake. See Appendix A for test capabilities and controls.

To characterize the sensitivity, accuracy, and specificity of the system, we used two T-cell lines, A3.01 (uninfected) and ACH-2 (HIV-infected), and two monocytic cell lines, HL-60 (uninfected) and OM-10 (HIV-infected). The infected cell line can be reactivated with PMA or TNF-α [46,47]. Additionally, HeLa cells (ATCC-CCL2) were used as a negative control. Thus, to quantify the sensitivity, accuracy, and specificity of the system, we diluted the uninfected (HL-60 or A3.01, 1 to 10 cells per assay or well) and HIV-infected cells (ACH-2 or OM-10, 1 to 10 cells per assay) into HeLa cells (10^3^ to 10^12^). Cells were pelleted and sectioned to identify infected cells. Then, we stained for HIV DNA, HIV-mRNA, and HIV-p24, as well as DAPI, Alu repeats (host DNA), and actin (cytoplasm) to quantify the numbers of cells with integrated HIV DNA, mRNA, and viral protein using microscopy with 10–20 individual assays per condition to obtain the percentage of positive cells as indicated (see Table 2, Table 3, Table 4 and Table 5 depending on the cell type analyzed: HL-60, A3.01, OM-10, or ACH-2 cells). 

Overall, our results indicate that diluting the uninfected cells (HL-60, or A3.01) into 10^3^ to 10^12^ uninfected HeLa cells does not result in the detection of HIV DNA, HIV-mRNA, or HIV-p24 as expected (Table 2 and Table 3). Furthermore, there was no detection of viral components in pure cultures of HeLa cells (data not shown). In contrast, the dilution of HIV-infected OM-10 or ACH-2 cells (myelocytic and T cells, respectively) into 10^3^ to 10^12^ uninfected HeLa cells demonstrated that HIV DNA, HIV-mRNA, and HIV-p24 detection was sensitive and accurate (Table 4 and Table 5). Upon treatment of the HIV-infected cells with PMA (5 nM) or TNF-α (50 U/mL) and subsequent dilution into HeLa cells, the reliability of detecting the viral mRNA and proteins was increased probably due to the higher expression of viral mRNA and proteins (data not show). Thus, our system was highly sensitive and specific to detect low numbers of HIV-infected cells with low amounts of HIV DNA, mRNA, and viral proteins among millions of uninfected cells. The distribution of the different viral factors staining for DAPI, HIV-nef DNA, viral mRNA, HIV-p24, and Alu repeats are presented in Appendix A. We denoted the specificity of each component in HIV-infected cells, OM-10 and ACH-2, and the absence of the viral components in uninfected cells HL-60 and A3.01 within Appendix A. Furthermore, to demonstrate the specificity of the staining, we calculated the Pearson’s colocalization of DAPI and Alu repeats, close to 0 to 100% (Appendix A, even though the coefficient number was 0 to 1). Only in HIV-infected cells, OM-10 and ACH-2, HIV DNA signal colocalized with Alu-repeat signals, but not in the uninfected cells. Similar numbers were observed for HIV DNA and DAPI signals. HIV DNA signal poorly colocalized with viral mRNA (19.66 ± 5.19 and 19.38 ± 3.70, for OM-10 and ACH-2 cells) and HIV-p24 (6.46 ± 2.38 and 9.16 ± 2.73, for OM-10 and ACH-2 cells) (see Appendix A). Overall, our system was highly sensitive and accurate in detecting low numbers of infected cells, as well as low amounts of HIV DNA, HIV-mRNA, and viral proteins.

## 3. Method Details

### 3.1. Materials

The slides contained brain tissue sections, xylene, ethanol, and a PNA ISH detection kit (Dako, cat. no. K5201, Santa Clara, CA, USA). Tris-buffered saline (TBS): 50 mM Tris/150 mM NaCl, pH 7.4, freshly prepared, peptide nucleic acid probe (PNA Bio, Newbury Park, CA, USA) for DNA detection, RNAscope 2.5 HD Detection Reagent-RED (ACD, cat. no. 322360, Newark, CA, USA) for RNA detection, antigen retrieval solution (Dako cat. no. S1700, Santa Clara, CA, USA), blocking solution (see recipe below), PNA Bio DNA probes for Alu repeats conjugated with Cy5 (Cy5-GCCTCCCAAAGTGCTGGGATTACAG), and *nef* conjugated with Alexa Fluor 488 (Alexa488-GCAGCTTCCTCATTGATGG). Primary antibodies: α-Iba-1 (Abcam, Cambridge, UK), α-GFAP (Sigma, Darmstadt, Germany), α-HIV p24 (Genetex, Irvine, CA, USA and the NIH AIDS repository), α-HIV nef, α-HIV tat, α-HIV integrase, α-HIV gp120, or α-HIV vpr (NIH AIDS repository). We used multiple antibodies thanks to the combined efforts with the NIH AIDS repository. The currently available antibodies are α-HIV p24 (Genetex, GTX40774, Irvine, CA, USA; NIH AIDS repository, HRP-20068, or ARP-4121), α-HIV nef (NIH AIDS repository, ARP-1124, 2949, 709), α-HIV tat (NIH AIDS repository, ARP-4672, or 466), α-HIV integrase (NIH AIDS repository, ARP-7375, or 3514), α-HIV gp120 (NIH AIDS repository, ARP-1476, 2534, 11682, or 11438), or α-HIV vpr (NIH AIDS repository, ARP-11836).

Secondary antibodies: α-goat Alexa Fluor 594, α-mouse Alexa Fluor 647, streptavidin Alexa Fluor 680, streptavidin Alexa Fluor 647, staining jars for slides, hydrophobic barrier pen ImmEdge (Vector Laboratories, cat. no. H-4000, Newark, CA, USA), HybEZ hybridization system (ACD, HybEZ 310010, Newark, CA, USA), and an orbital shaker at 55 °C.

### 3.2. Staining Methods

**Rehydration of samples:** Paraffin-embedded slides containing the tissue samples were immersed in the following solutions consecutively: xylene for 5 min (2 times), 100% EtOH for 3 min, 100% EtOH for 3 min, 95% EtOH for 3 min, 90% EtOH for 3 min, 70% EtOH for 3 min, 60% EtOH for 3 min, 50% EtOH for 3 min, and miliQ H_2_O for 3 min. Then, tissue was encircled with ImmEdge Pen to reduce the reagent volume needed to cover the specimens. Finally, the slides were immersed in miliQ H_2_O for 3 min. As described below, archival samples needed additional steps to eliminate autofluorescence and antigen retrieval. However, the protocol of staining worked for most slices. It was impossible to detect infected cells only in the lymph nodes and spleen obtained from HIV-infected individuals archived for more than 20 years with an unknown black precipitant in the tissue. The nature of the black precipitate was unknown and interfered with fluorescence detection.

**Pre-treatment with proteinase K:** Tissues were incubated with proteinase K diluted at 1:10 in 1X TBS (PNA ISH kit, Newark, CA, USA) for 10 min at RT in a humidity chamber. Slides were immersed in miliQ H_2_O for 3 min, then immersed in 95% EtOH for 20 s, and finally, the slides were air-dried for 5 min.

**HIV DNA probe hybridization:** Tissues were incubated with a PNA DNA probe 10 µM for *nef*-PNA Alexa Fluor 488 and Alu-PNA Cy5. Slides were placed in a pre-warmed humidity chamber and incubated at 42 °C for 30 min; then, the temperature was raised to 55 °C for an additional 1 h of incubation. Subsequently, tissues were incubated using preheated Stringent Wash working solution (PNA ISH kit, Newark, CA, USA) diluted at 1:60 in TBS for 25 min in an orbital shaker at 55 °C. Slides were equilibrated to RT by brief immersion in TBS for 20 s, as described in HIV and SARS-CoV-2 samples [31,42,43,48].

**HIV mRNA detection:** The manufacturer followed RNAscope 2.5 HD Detection Reagent-RED protocol. A probe for HIV-gag-pol mRNA was added to the tissue samples and incubated for 30 min at 42 °C, and then 50 min at 55 °C. Samples were incubated in Preheat Stringent Wash working solution diluted at 1:60 in TBS (PNA ISH kit, Newark, CA, USA) for 15 min in an orbital shaker at 55 °C. Slides were immersed in TBS for 20 s. The colorimetric method was performed as described by the manufacturer. The development of the viral mRNA signal needed to be low, so we did not overdevelop it. The reaction was stopped as soon as a red color was observed. Samples were washed three times with miliQ H_2_O. Slides should be kept at 4 °C in the dark to minimize fluorescence decay. The detection of RNA scope by color was not reliable for detecting viral reservoirs with low copies of viral RNA, especially for archival or tissues in formalin or other fixatives for long periods. Thus, the detection of viral mRNA was achieved by fluorescence instead of colorimetric analysis.

**HIV or cellular protein detection:** We recommend these treatments for all samples. Antigen retrieval was performed by incubating slide sections in a commercial antigen retrieval solution (Dako, Santa Clara, CA, USA or homemade) for 30 min in a water bath with a temperature close to 80 °C. Slides were removed from the bath and then allowed to cool down in TBS. Samples were permeabilized with 0.1% Triton X-100 for 2 min and then washed in TBS for 5 min 3 times. Unspecific antibody binding sites were blocked by incubating samples with a freshly prepared blocking solution. Sections were incubated overnight at 4 °C using a humidity chamber (10 mL of blocking solution: 1 mL 0.5 M EDTA, 100 μL fish gelatin from cold water 45%, 0.1 g albumin from bovine serum fraction V, 100 μL horse serum, 500 μL human serum, and 9 mL of PBS). A primary antibody was added to the samples diluted in blocking solution and incubated at 4 °C overnight. Slides were washed in PBS 5 min 3 times to eliminate unbound antibodies. Secondary antibodies were added at the appropriate dilutions and incubated for 2 h at RT. Slides were washed in PBS for 5 min 3 times to eliminate unbound antibodies. Slides were mounted using Prolong Diamond Antifade Mountant containing DAPI. Slides were kept in the dark at 4 °C.

**Fixation of Tissues and Cells:** To identify viral reservoirs and residual viral replication, we used several tissue types (urethra, vaginal, brain, spleen, liver, bone marrow, lung, heart, and gut), species (macaques and humanized mice), time of fixation (days, months, and years), type of fixation (formalin, Bouin, PFA, and alcohol/acid-related), the thickness of the section (10 to 300 µm), and tissue compromise (for example, in COVID-19 cases the tissue structure is altered and significant debris and red blood cell lysis occur compromising the red/green channels). Fixation in ethanol (cold 70%, −20 °C for 20 min) or PFA is preferable for cells.

**Antigen retrieval**: There are several techniques for antigen retrieval (www.ihcworld.com/epitope_retrieval.htm, accessed on 29 June 2022), and most work perfectly for the applications indicated below. We used the boiling citrate buffer method for 15 min (pH 6.0) for thicker tissue sections (10 to 300 µm) for our applications, but we also obtained good results with microwave-based techniques.

**Elimination of autofluorescence**: (a) Natural autofluorescence due to flavins, porphyrins, and chlorophyll is common. If fixation is performed most of the time, this autofluorescence is concentrated in areas. Additionally, these auto-fluorescent host compounds can be redistributed during cutting and solvent treatments, resulting in background fluorescence. However, new optical configurations in most microscopes provide unmixing and/or spectral detection to detect and reduce this problem. In addition, treatment of the tissues with Sudan Back (0.3% in 70% ethanol) stirred in the dark for 2 h will significantly reduce the autofluorescence produced by flavins, porphyrins, and chlorophyll. (b) Another artifact to be aware of is the accumulation of elastin and collagen (both produce autofluorescence). Elastin contains several potential fluorophores when there is a cross-linking of tricarboxylic amino acid with a pyridinium ring [49,50]. The detection of these products is minimal in small vessels, but it is a significant problem in large vessels. To eliminate autofluorescence from elastin products, incubate the samples in 0.5% pontamine sky blue and 6.6′-[(3,3′-dimethoxy[1,1′-biphenyl]-4,4′-diyl)bis(azo)]bis[4-amino-5-hydroxy-1,3-naphthalenedisulfonic acid]; tetrasodium salt dissolved in 50 mM Tris buffer before mounting the samples. However, this sometimes compromises the red channel detection of fluorophores. If the red channel needs it, an alternative solution is adding 0.1% toluidine blue for 3 min before mounting the samples, but this does not work in all vessels. The interpretation of the fluorescence must be cautious. (c) For archival samples, aldehydes react with amines and proteins to generate fluorescent products, especially in samples incubated for a long time in fixatives. This problem occurs most often in fixatives, such as glutaraldehyde and formaldehyde. For tissue sections 10 to 300 µm, we incubated them 5 times for 15 min each in a solution of fresh borohydride (1 mg/mL dissolved in PBS and prepared on ice). After this process, we washed in PBS 3 times and discarded the leftover sodium borohydride. (d) To reduce overall autofluorescence, we exposed the tissue slices to fluorescent lights in each channel. No company currently sells the appropriate lightboxes, but they are relatively easy to construct. To build a specific wavelength lightbox (similar to the one used to detect ethidium bromide in agarose gels), fluorescent tubes, especially for blue, green, red, and far-red channels, can be purchased from several companies. These can be used to “burn” the autofluorescence in the tissue sections before the staining process. All these protocols need to be calibrated for each application. In the cases of the present study, lightbox and sodium borohydride were sufficient.

**Endogenous biotin blocking**: In our analysis, probes with biotin could be used. In that case, endogenous biotin blocking is suggested using a biotin-blocking kit. First, incubate the tissue sections or cells with avidin solution for 10–30 min and then wash in TBST 3 times for 5 min. Next, incubate with biotin solution for 10–30 min and then wash in TBST 3 times for 5 min.

**Antibody conjugation and pre-absorption:** Currently, few HIV antibodies have been calibrated or validated to detect low levels of viral protein per cell and/or within a few cells, including gp120 and HIV-p24 [33,42,43,44,51]. However, most antibodies for gp120, integrase, nef, vpr, vpu, and tat are poorly immunogenic and unspecific, complicating the interpretation of the results, especially in tissues. Thus, multiple controls are essential to achieve proper viral protein identification. In collaboration with the NIH repository and their authorities, we obtained 10–15 tubes of each antibody (several batches and catalog numbers to test multiple ones) to purify, preabsorb, and conjugate them with biotin to amplify the specific signals.

We chose biotin because it is a rapid and specific way to conjugate the antibody and remove endogenous biotin from tissues. In addition, biotin is resistant to extreme conditions, such as heat, pH, chemicals, and proteolysis, all the conditions described above to reduce autofluorescence. Additionally, biotin is an easy way to amply your signal-to-noise ratio due to the fact that 3–6 biotin molecules are conjugated to each antibody.

To purify, characterize, and examine antibody specificity, we first obtained a significant stock of antibodies to concentrate and purify them before labeling; second, significant amounts of uninfected tissues (in this case, brains) to pre-absorb the anti-HIV protein antibodies; third, good tissue and cell control in the presence and absence of HIV (see calibration of the staining using cell lines and uninfected HeLa cells, see Appendix A); fourth, good-quality uninfected and HIV-infected tissues with high, medium, and low viral protein expression; and lastly, a good judgment to evaluate controls, signal localization, and equipment/program calibration.

As described above, we concentrated and purified the anti-HIV antibodies (monoclonal and polyclonal) according to the isotype and species (1–4 mg/mL). A great source used to decide on the purification strategy can be found via the following link: https://tools.thermofisher.com/content/sfs/brochures/1601975-Antibody-Production-Purification-Guide.pdf (accessed 25 July 2022). After the purification, we biotinylated the antibodies as described above. Before testing, all biotinylated antibodies were tested in uninfected and HIV-infected cell lines as described. Here, we evaluated specificity by competing the purified antibody binding with the specific recombinant proteins (100–300 µg/mL). Moreover, if the purified antibodies had unspecific backgrounds or unrelated signals, the antibodies were not used for subsequent experiments. We eliminated several options from commercial and government repositories due to unspecific signals.

After purification and antibody biotinylation, we preabsorbed the antibodies in uninfected human brain tissue lysate (n = 3 different individuals, 50–150 µg per mg of antibody). We repeated the above protocol to repurify and test the antibodies in uninfected and HIV-infected cells, competitive experiments with recombinant proteins using uninfected and HIV-infected tissues, as well as uninfected and HIV-infected cell lines. Additionally, controls using tissues from humanized animal models (uninfected and HIV infected) were analyzed to determine antibody specificity. Most antibodies to gp41, p17, tat, gp120, nef, reverse transcriptase, vif, vpr, and vpu did not work, or the signal “vanished” after pre-absorption in the uninfected human brain or was not affected by the use of the respective recombinant protein to compete for the specific binding. As a positive control for our purification and validation, antibodies to human CXCR4 and CCR5 were included in the analysis (except for pre-absorption). After these controls were properly passed, we moved into the analysis of human uninfected and HIV-infected tissues as described in the current communication. Below, we provide a detailed protocol for staining. Each section can be performed independently.

### 3.3. Detailed DNA/mRNA/Antibody Staining

Samples were fixed and prepared as described above (antigen retrieval and elimination of autofluorescence).The sections were deparaffinized as described above using sequential alcohols and xylenes.Afterward, either antigen retrieval or endogenous biotin blocking was performed as described above. The brain generally has low to undetectable levels of endogenous biotin. However, liver, spleen, and lymph node tissues had significant endogenous biotin amounts.After HIV DNA and mRNA staining, tissue sections or cells were blocked overnight using a blocking solution containing 50 mM EDTA, 1% horse serum, 1% Ig-free BSA, 5% human serum, and 1% fish gelatin in PBS.Samples were incubated overnight in the primary antibody at 4 °C (HIV and cell marker antibodies). A critical point was determining how many antibodies can be used concomitantly based on antibody species and isotypes. Several combinations, including species, isotypes, labeling, and secondary antibodies, can be used to avoid repeating the same host.

A critical point of the experimental approach was to use appropriate negative controls, including non-immune purified IgGs (the same concentration as the HIV antibodies), rabbit serum, or rabbit purified IgG (the same concentration as the serum or immune IgG) and non-immune goat serum or IgG (the same concentration as Iba-1 IgGs). More importantly, negative controls only using primary or secondary antibodies were inaccurate and compromised the detection because Fc binding was not properly examined. To some degree, all the cells, tissues, and fluids had non-specific binding that was necessary to consider, especially in the cases of the detection of low amounts of proteins, such as those found in HIV reservoirs. Thus, the specificity of the antibodies must be confirmed by replacing the primary antibody with the appropriate isotype-matched control reagent, anti-IgG_1_, IgG_2A_, IgG_2B_, or the IgG fraction of normal rabbit/mouse/goat/Armenian hamster/chicken or other serums depending on the primary antibody being used.

F.After incubation with the primary antibodies, at least 5 washes with PBS every 10 min were required to eliminate the unbounded antibodies.G.To detect the antibodies conjugated to biotin, incubation with streptavidin-conjugated to a fluorophore was necessary. An additional control was required here: infected tissues without primary biotinylated antibodies but with streptavidin conjugated to a fluorophore, to examine whether the endogenous biotin was properly eliminated. The detection of low levels of HIV proteins required at least 3 h of incubation with the labeled streptavidin or secondary antibody.H.After incubation with the secondary antibody or conjugated streptavidin, at least 5 washes with PBS were required to eliminate the unbounded streptavidin or secondary labeling antibodies. Detection was achieved using confocal equipment equipped with unmixing and spectrum-detection systems that enabled the separation of extremely narrow wavelengths (up to 2.5 nm) to identify and isolate multiple colors (wavelengths) without overlay or unspecific signals (see examples in [31,33,43,44,48]).I.Then, samples were mounted using Prolong Gold/Diamond anti-fade reagent with DAPI or Vector Shield (preferred).J.After staining, samples were kept in the dark before confocal analysis to enable the mounting media to penetrate the tissue and allow for the media to cure.

**Data acquisition and analysis**. The three-dimensional reconstruction and deconvolution of areas of interest were performed from 6 to 8 optical sections obtained at 0.5 or 0.95 μm intervals in large pieces of tissue. Upon identifying the HIV DNA-positive areas with colocalization with DAPI and Alu-repeat staining, the tissue coordinates were selected and rescreened at better resolutions; 12 to 25 optical sections were obtained at 0.250 or 0.150 μm intervals. However, most of the brain tissue was negative for HIV components. Thus, the figures presented are the selected positive areas. To analyze and quantify the abundance and colocalization of the probes and antibodies, the number of positive pixels and their intensity in microglia/macrophages or astrocytes were measured in specific regions of interest, and the Pearson’s and Mander’s colocalization coefficients were calculated. Similar numbers of cells and areas were included in the 3D deconvoluted optical reconstructions to compare the abundance of signals across the brain. This analysis avoided problems associated with inflammatory versus normal tissues where the density and structure of the tissue were different. Adjacent sections were stained with hematoxylin and eosin (H&E); we correlated the confocal findings with histopathology of the same tissue sections. Colocalization and numbers of positive pixels for each color were quantified using NIS-Elements imaging software (Nikon, Japan). The tissues analyzed were mostly the frontal cortex and some hippocampal, as indicated in Table 1.

**Neurocognitive examination**. NNTC and CHARTER perform a comprehensive neurocognitive test battery every 6 months, including motor function (perceptual-motor speed), verbal fluency, executive function, attention/working memory, speed of information processing, learning, and memory; for details, see http://www.mountsinai.org/patient-care/service-areas/neurology/areas-of-care/neuroaids-program or https://nntc.org/ (both accessed 25 July 2022) [52,53,54].

**Statistical analysis.** Statistical analyses were performed using Prism 5.0 software (GraphPad Software, Inc., San Diego, CA, USA). Analysis of variance was used to compare the different groups; * *p* ≤ 0.005 compared to uninfected conditions, # *p* ≤ 0.005 compared to HIV_un_ conditions, and & *p* ≤ 0.005 compared to HIV_un_ or HIV_low_ conditions; n = 34 tissues were analyzed, and 21 tissues were compared to uninfected tissues, Un and Alz, n = 8 different tissues; each point represents 3–5 different areas per tissue analyzed. To compare the different groups, two-way ANOVA was used. ANOVA, as indicated in the text, revealed a significant difference with F-values higher than 70.54, *p* < 0.0001, η^2^ = 24.70%.

## 4. Results

**Detection of viral reservoirs in human brain tissue samples.** In collaboration with the National NeuroAIDS Tissue Consortium (NNTC) and Neurobiobank, we collected 34 brain tissue samples from uninfected, HIV-infected, and uninfected patients with Alzheimer’s disease (cortex and/or hippocampus) (see Table 1 for details). There were no significant differences in age between HIV-negative and HIV-positive groups (HIV-positive, mean = 49.9 ± 2.6 years; HIV-negative, mean = 50.4 ± 5.2 years; Table 1). Sex corresponded to 38% female and 62% male for the HIV-infected population and 20% female and 80% male for the uninfected population (Table 1). The HIV-positive cohort had an average of 14.9 ± 1.9 years living with HIV, which we could subdivide into three groups according to systemic viral replication upon demise: (a) HIV-positive patients with undetectable replication (patients N° 6–13, HIV_un_) with a range of plasma HIV RNA of 0 to <50 copies/mL and an average CD4^+^ T-cell count of 727 ± 177 cells/mm^3^; (b) low replication (patients N° 14–19, HIV_low_) with a range of plasma HIV RNA of 275 to 12,352 copies/mL and an average CD4^+^ T-cell count of 444 ± 147 cells/mm^3^; and high replication (patients N° 20–26, HIV_high_) with a range of plasma HIV RNA of 165,862 to >750,000 copies/mL and an average CD4^+^ T-cell count of 22.4 ± 12.4 cells/mm^3^ (see Table 1 for details). Among the HIV-positive participants, 72% had some degree of cognitive impairment as determined by neuropsychological tests (Table 1). Additionally, we included archival cases of AIDS and encephalitis (HIVE, patients N^o^ 27–31) as a positive control and acute cases of Alzheimer’s disease as a negative control (Alz, patients N^o^ 32–34). To assure an unbiased assessment, all samples were received and analyzed blindly. After all the data were acquired, the clinical data and HIV status were requested to assure proper scientific rigor.

Brain cells containing integrated HIV DNA accumulate in cell clusters containing microglia/macrophages and astrocytes in the current ART era. To identify the viral reservoirs containing HIV DNA in the nucleus within the human brain, we used a combination of tissue staining, confocal microscopy, 3D reconstruction, deconvolution, and image analysis. A graphical example of the approach is presented in Figure 1A, and the protocol is described in detail in the Methods Section. Large brain tissue pieces were serially sectioned to characterize viral reservoirs (cortex and hippocampal sections’ sizes were 6.7 ± 3.03 cm^2^, 11 to 12 serial sections of 10 µm each). The reason for using large tissue pieces was the low abundance of viral reservoirs in some tissues. An example of our approach was that the first and last sections were used for hematoxylin and eosin (H&E); the second section for trichrome staining; and the subsequent sections were stained for DAPI to identify nuclei, HIV DNA probe for the *nef* DNA sequence (HIV-*nef-DNA* or HIV DNA), HIV-mRNA probe for HIV-gag-pol mRNA sequence (HIV-mRNA or mRNA^gagpol^), Alu-repeats probe as control of host DNA (Alu), and HIV-p24 protein (HIV-p24 prot or protein^p24^) to quantify the overall number of cells with integrated DNA and positive for viral mRNA and HIV-p24 proteins versus the total number of cells (Alu repeats and DAPI positive). The following section was stained using DAPI for nuclei, HIV DNA probe for *nef*, Iba-1 protein to quantify microglia/macrophages, and GFAP to quantify astrocytes. Upon identifying particular cell types with integrated HIV DNA, the subsequent tissue section was used to identify viral protein expression in particular cell types. After nine sections, we repeated the staining for the HIV DNA probe for *nef*, Alu-repeats probe, and DAPI to reposition the brain structures, clusters of HIV-infected cells, and cell types identified (Figure 1A).

Using the staining and imaging approaches described above, we identified, quantified, and characterized low-abundance HIV components (HIV DNA, mRNA, and proteins as described in Figure 1A) and the cell type positive for each component in the human brain obtained from individuals under ART. We identified that most cells with integrated HIV DNA (triple-positive nucleus staining for DAPI, Alu repeats, and the HIV DNA probe) were organized in cell clusters of macrophages and astrocyte cells (Figure 1B–G). The analysis of brain tissue sections obtained from uninfected or Alzheimer’s individuals shows DAPI, Alu repeats, and cellular markers (Iba-1 and GFAP) and no unspecific staining for HIV DNA, mRNA, or HIV-p24 protein (Appendix A for large brain areas using tissues; the examples correspond to tissues of 6.7 ± 3.03 cm^2^. Most areas are negative, but the areas containing HIV reservoirs are shown, see Bar: 200 µm, from uninfected, A-K, HIV-infected individuals with undetectable systemic replication, HIV_un_, L-V, and HIV-infected individuals with encephalitis, HIVE, and W-G1). In brain tissues obtained from HIV-infected individuals, cell clusters containing HIV DNA staining were positive for the microglia/macrophage marker, Iba-1 protein, and surrounded by enlarged hyper-reactive GFAP-positive astrocytes (Figure 1B–G, a representative example). The number of HIV DNA-positive clusters per tissue section corresponded to a total of 2.67 ± 1.2, 5.38 ± 1.1, and 22.40 ± 10.09 in brain tissues obtained from HIV-infected individuals with undetectable (HIV_un_), low (HIV_low_), and high (HIV_high_) replication, respectively (Figure 1H; * *p* ≤ 0.005 compared to uninfected conditions, # *p* ≤ 0.005 compared to HIV_un_ conditions, and & *p* ≤ 0.005 compared to HIV_un_ or HIV_low_ conditions; n = 34 tissues analyzed and 21 tissues compared to uninfected tissues, Un and Alz, n = 8 different tissues; each point represents 3–5 different areas per tissue analyzed). In HIVE brains, 46.83 ± 14.74 cell clusters were detected (Figure 1H; * *p* ≤ 0.005 compared to uninfected conditions, # *p* ≤ 0.005 compared to HIV_un_ conditions, and & *p* ≤ 0.005 compared to HIV_un_ or HIV_low_ conditions, n = 5 HIVE tissues compared to uninfected tissues, n = 8; each point represents 3–5 different areas per tissue analyzed). HIVE conditions correspond to archival tissues from the early stages of the AIDS epidemic without ART or only on monotherapy. We determined that the extent of HIV systemic replication (see Table 1, more than three years) could predict the numbers of clusters with HIV-infected cells (Figure 1H, compare HIV_un_, HIV_low_, and HIV_high_ or HIVE). We must denote that most large tissue areas were negative for HIV DNA, and the examples in the figures focus on the positive areas. Additionally, in HIV_un_, HIV_low_, and HIV_high_ conditions, there was a correlation number of HIV-positive clusters and ART status, suggesting that long-term ART reduces the number of clusters of HIV-infected cells (Figure 1H, F_value_ = 222.97, ANOVA). No HIV DNA staining was detected in uninfected tissues (brains obtained from uninfected, Un, and Alzheimer’s patients, Alz; Figure 1H and see Appendix A). In addition, we could not detect HIV reservoirs using Alu-PCR or ultra-sensitive PCR, as described [12,14], using the brain tissues obtained from individuals with undetectable and low replication. Moreover, Alu-PCR and ultra-sensitive PCR detected viral reservoirs in brain tissues obtained from patients with high systemic HIV replication (HIV_high_) and HIVE tissues with high reliability rather than samples from patients with low or undetectable replication (HIV_un_, HIV_low_), where results were variable and inconclusive (data not shown).

Most cell clusters positive for HIV DNA signal were also positive for HIV-mRNA, 0.86 ± 0.51 for HIV_un_, 1.88 ± 0.52 for HIV_low_, and 14.22 ± 8.3 for HIV_high,_ indicating that approximately 1/3 of the cells containing HIV DNA could express viral mRNA (Figure 1I; * *p* ≤ 0.005 compared to uninfected conditions, # *p* ≤ 0.005 compared to HIV_un_ conditions, and & *p* ≤ 0.005 compared to HIV_un_ or HIV_low_ conditions; n = 34 tissues analyzed and 21 tissues compared to uninfected tissues, Un and Alz, n = 8 different tissues; each point represents 3–5 different areas per tissue analyzed). In contrast, in HIVE conditions, the cell numbers containing HIV DNA matched the cells expressing HIV-mRNA supporting highly productive infection (Figure 1I; HIVE). Further analysis of the cell clusters for HIV-p24 protein expression indicated 0.37 ± 0.32 for HIV_un_, 1.16 ± 0.37 for HIV_low_, and 9.88 ± 6.9 for HIV_high,_ indicating that approximately 1/3 of the cells containing HIV DNA and HIV-mRNA for each group still expressed HIV-p24 in these cell clusters (Figure 1J; * *p* ≤ 0.005 compared to uninfected conditions, # *p* ≤ 0.005 compared to HIV_un_ conditions, and & *p* ≤ 0.005 compared to HIV_un_ or HIV_low_ conditions; n = 34 tissues analyzed and 21 tissues compared to uninfected tissues, Un and Alz, n = 8 different tissues; each point represents 3–5 different areas per tissue analyzed). However, if we compared HIV_high_ with HIV_low_ and HIV_un_ groups, the HIV DNA-positive clusters decreased by at least ~75% upon ART (Figure 1). These data clearly denote the positive effects of ART on the HIV brain reservoir cell population. In HIVE conditions, HIV-p24 expression correlates with HIV DNA and HIV-mRNA expression (Figure 1J; HIVE).

Overall, in the ART era, the number of clusters of HIV-infected tissues decreased by about 2/3 compared to HIV_high_ and HIVE conditions (Figure 1H) and confirmed some of the ratios described for T-cell reservoirs [12,15,16]. Furthermore, there was a decrease in the overall intensity of HIV-mRNA and HIV-p24 signals between low-replication (HIV_un_ and HIV_low_) and high-replication groups (HIV_high_ and HIVE) of 78.34 ± 20.41% for HIV-mRNA and 83.04 ± 16.54% for HIV-p24 (Figure 1I,J), indicating that in the ART era, viral mRNA and protein were expressed at lower levels (data not represented, *p* ≤ 0.001 compared to HIVE conditions, respectively). The analysis of brain tissue sections obtained from uninfected individuals (uninfected, Un, and Alzheimer’s disease, Alz) showed DAPI, Alu repeats, and cellular markers (Iba-1 and GFAP), but no unspecific staining for HIV DNA, HIV-mRNA, or HIV-p24 as expected (Appendix A). Moreover, there was no correlation between the numbers of cell clusters, numbers of clusters per tissue, and the ratio among HIV DNA/HIV-mRNA/HIV-p24 with the degree of cognitive decline in the population analyzed (HIV-associated minor cognitive/motor disorder, MCMD or HIV-associated dementia, HAD data not represented). In conclusion, long-term ART and the reduction in systemic replication reduced the numbers of HIV-infected clusters in the brain, but it also became evident that a significant number of HIV-infected cell clusters remained in the human brain producing residual viral mRNA and HIV-p24 protein.

**Brain cells, microglia/macrophages, and astrocytes containing integrated HIV DNA still produce low levels of HIV-mRNA and HIV-p24.** Subsequently, we expanded our analysis to larger cortical and subcortical areas to quantify viral components’ frequencies within different cell types (see details in Table 1, cortex and hippocampus). Representative images of unstained and H&E-stained conditions (Figure 2A,B, respectively). The blue boxes in both figures denote the presence of clusters with HIV-infected cells (HIV DNA positive cells; Figure 2A,B). Upon identification of the coordinates with HIV DNA staining and colocalization with Alu repeats and DAPI, we amplified the selected areas to examine the expression and distribution of DAPI (nuclear marker; Figure 2C), HIV DNA (Figure 2D; HIV-*nef*-DNA), gag-pol mRNA (HIV-mRNA; Figure 2E), HIV-p24 (Figure 2F), Alu repeats (Figure 2G), and the merge/colocalization of all colors (Figure 2H; merge) as described in the Methods Section. To quantify and identify the cell types that contained integrated HIV DNA and the degree of viral mRNA and protein expression, we quantified the HIV DNA signal in a cell-specific manner (Iba-1 or GFAP-positive cells to identify microglia/macrophages or astrocytes, respectively) in the different tissues analyzed (see Table 1).

In brain tissue samples obtained from HIV-infected individuals with undetectable replication (HIV_un_; Figure 2I), microglia/macrophages corresponded to 21.48 ± 13.50 and astrocytes corresponded to 74.85 ± 15.85 cells with HIV DNA per tissue analyzed (Figure 2I; HIV_un_). In brain tissues obtained from HIV-infected individuals with low systemic replication, 131.1 ± 28.17 cells corresponded to microglia/macrophages and 76.73 ± 16.11 cells corresponded to astrocytes (Figure 2I; HIV_low_). In brains obtained from HIV-infected individuals with high systemic replication, most HIV DNA-positive cells corresponded to microglia/macrophages, 307.23 ± 87.73 cells, and 44.43 ± 19.11 cells were astrocytes (Figure 2I; HIV_high,_ * *p* ≤ 0.005 compared to uninfected conditions, # *p* ≤ 0.005 compared to HIV_un_ conditions, and & *p* ≤ 0.005 compared to HIV_un_ or HIV_low_ conditions; n = 34 tissues analyzed and 21 tissues compared to uninfected tissues, Un and Alz, n = 8 different tissues; each point represents 3–5 different areas per tissue analyzed).

Our data indicate that microglia/macrophages are the predominant cells infected with HIV in patients with low to high systemic replication (Figure 2I; HIV_low_ and HIV_high_). However, in HIV conditions with undetectable replication, the number of HIV-positive microglia/macrophages decreased significantly (Figure 2I; HIV_un_, F = 287.53 for ANOVA to compare all the groups). In contrast, in GFAP-positive cells (astrocytes), the numbers of astrocytes containing HIV DNA in the nucleus were not altered by systemic replication (Figure 2I; HIV_un_, HIV_low_, and HIV_high_; no significant differences). Overall, our data indicate that the HIV-infected astrocyte population is less sensitive to long-term ART than the myeloid-infected population (Figure 2I; compare microglia/macrophages and astrocytes).

To determine the expression of viral mRNA within the cells containing HIV DNA, we quantified the numbers of HIV-mRNA and HIV DNA-positive cells. Analyses of brains obtained from HIV-infected individuals and stained for Iba-1 indicated that only 5.85 ± 2.9 cells per tissue expressed viral mRNA (Figure 2J; HIV_un_), despite the fact that 21.48 ± 13.49 cells per tissue showed viral DNA in the nucleus (compare Figure 2I,J, for the quantification of HIV DNA containing cells). In brain tissues obtained from individuals with low systemic HIV replication, only 47.65 ± 20.02 cells per tissue analyzed were positive for HIV DNA and viral mRNA (Figure 2J; HIV_low_). In contrast, in brain tissues obtained from individuals with high viral replication, the population of microglia/macrophages cells positive for HIV DNA and viral mRNA was similar (HIV DNA, 307.22 ± 87.7 versus 224.32 ± 106.42, viral mRNA), suggesting that effective ART reduces the replication of the virus in myeloid cells (Figure 2J; * *p* ≤ 0.005 compared to uninfected conditions, # *p* ≤ 0.005 compared to HIV_un_ conditions, and & *p* ≤ 0.005 compared to HIV_un_ or HIV_low_ conditions; n = 34 tissues analyzed and 21 tissues compared to uninfected tissues, Un and Alz, n = 8 different tissues; each point represents 3–5 different areas per tissue analyzed).

In contrast, the quantification of viral mRNA expression in HIV DNA-positive astrocytes indicated the fact that, despite different levels of HIV replication, viral mRNA expression was maintained: 55.45 ± 15.16 HIV_un_, 62.63 ± 16.69 HIV_low_, and 40.65 ± 18.31 HIV_high_ (Figure 2J; * *p* ≤ 0.005 compared to uninfected conditions and significant differences between the HIV-infected groups. ANOVA analysis among the HIV-infected groups was not significant), suggesting that astrocytes were less sensitive to ART than myeloid cells.

To determine whether cells containing HIV DNA and viral mRNA could produce viral proteins, we first quantified the expression of HIV-p24. The quantification of triple-positive cells, HIV DNA, HIV-mRNA, and HIV-p24 signals in microglia/macrophages indicated that in the brain obtained from individuals with undetectable replication, the numbers of cells per tissue were analyzed 1.68 ± 0.94 cells (Figure 2K; HIV_un_). The analysis of brains from individuals with low systemic replication indicated that 32.7 ± 22.6 of microglia/macrophages (Figure 2K; HIV_low_) were triple-positive cells. Lastly, the analysis of brains obtained from HIV-infected individuals with high systemic viral replication indicated that 202.22 ± 101.52 of microglia/macrophages were triple-positive cells (Figure 2K; HIV_high_, * *p* ≤ 0.005 compared to uninfected conditions, # *p* ≤ 0.005 compared to HIV_un_ conditions, and & *p* ≤ 0.005 compared to HIV_un_ or HIV_low_ conditions; n = 34 tissues analyzed and 21 tissues compared to uninfected tissues, Un and Alz, n = 8 different tissues; each point represents 3–5 different areas per tissue analyzed).

These data indicate that despite long-term ART and effective systemic viral suppression, a population of microglia/macrophages in the brain still expresses HIV-p24 (Figure 2K). In contrast, in astrocytes containing HIV DNA and viral mRNA, a conserved HIV-p24 expression, despite the differences in systemic replication, was observed, HIV_un_, 42.58 ± 11.79 cells; HIV_low_, 37.1 ± 17.68; and HIV_high_, 37.92 ± 17.47 cells, indicating that viral replication and HIV-p24 production in astrocytes are stable and not affected by ART (Figure 2K). In contrast, the analysis of astrocytes for triple-positive cells (HIV DNA/viral RNA/HIV-p24) indicated that HIV-infected astrocytes were not sensitive to systemic replication due to the fact the expression of HIV-p24 remained equal under all the conditions analyzed (Figure 2K; astrocytes). Overall, we demonstrated that, first, even in the current ART era, viral reservoirs were present in the brain in microglia/macrophages and astrocytes; second, HIV-infected microglia/macrophages, but not astrocytes, were susceptible to long-term ART reducing viral or protein production within the brain (HIV DNA/mRNA and HIV-p24); third, ART reduced residual viral replication or viral protein production in HIV-infected microglia/macrophages but not in HIV-infected astrocytes; and, lastly, HIV-infected astrocytes were a more stable viral reservoir than microglia/macrophages after long-term infection and ART.

To compare the size of the current viral reservoir under long-term ART to the early stages of the AIDS epidemic, we used HIV encephalic tissues (see Table 1 for patient information) to quantify infection and cell numbers per tissue. First, HIV infection (HIV DNA, mRNA, and protein) in HIVE tissues was more widespread than the localized infection in clusters observed in the HIV_un_, HIV_low,_ and HIV_high_ analyzed (see Appendix A, HIVE, and Table 1). Second, the total number of HIV DNA cells corresponded to 1606 ± 368 cells in a comparable tissue volume analyzed (compared to Figure 2L). The HIV-infected population was mostly HIV-infected microglia/macrophages (945 ± 305 cells) and a small astrocyte population (430 ± 208 cells). Thus, a significant difference in the numbers and ratios between microglia/macrophages and astrocytes can be observed between current ART and HIVE conditions.

A critical piece of data obtained from our analysis of tissues from HIV-infected individuals with undetectable, low, and high systemic replication is the lack of colocalization of HIV-p24 protein with cells containing integrated HIV DNA or viral mRNA, but next to clusters of HIV-infected cells suggesting local secretion and bystander uptake. Most of the tissue was negative for viral markers, and the staining was highly localized in the HIV-infected clusters and surrounding cells for all the viral proteins examined. Our results indicate that most cells with HIV DNA and gag-pol mRNA express HIV-p24 (Figure 2L; HIV DNA (+), combined numbers for HIV_un_, HIV_low_, and HIV_high_); however, HIV-p24 protein was found in neighboring uninfected cells (452.12 ± 113.64 cells, Figure 2L, HIV DNA (−), * *p* ≤ 0.005 compared to uninfected conditions and # *p* ≤ 0.005 compared to HIV DNA (+) conditions). In contrast, the analysis of brain tissues obtained from the early and pre-ART era with encephalitis (HIVE) demonstrated that most HIV-p24 proteins were localized in HIV DNA-positive cells (HIVE DNA (+)). Low secretion into uninfected cells was detected (Figure 2L; HIVE DNA (+), 1497 ± 459 total cells, or HIVE DNA (−), 890 ± 278 total cells, * *p* ≤ 0.005 compared to uninfected conditions). We did not detect any unspecific staining for HIV DNA, viral mRNA, or HIV-p24 in uninfected tissues, Un, or tissues obtained from individuals with Alzheimer’s disease (see Appendix A). Overall, our data indicate that HIV-p24 is still produced by latently infected cells and is taken up by neighboring cells, even in the current ART era.

**HIV-gp120 is synthesized and released from HIV-infected cells and accumulates in neighboring uninfected cells even in the ART era.** Several studies have described the toxic effects of soluble viral proteins within the CNS, including gp120, integrase, nef, vpr, and tat, but conclusive data about production, release, and associated bystander toxicity are lacking under the current long-term ART era [55,56,57,58,59]. The gp120 protein is an envelope protein, and it is required for HIV entry into cells and is neurotoxic if applied as a soluble protein [60,61]. Furthermore, several animal models have been developed to examine the toxic effects of these viral proteins, but it is unclear whether gp120 is produced and secreted in the ART era compared to the early stages of the AIDS epidemic and its significant role in HIV-associated cognitive impairment [62,63,64].

Overall, HIV-gp120 distribution was similar to HIV-p24; most brain tissue samples were negative for HIV-gp120; however, HIV-gp120 expression was associated with specific cell clusters positive for the HIV DNA signal. A representative cell cluster is shown (Figure 3A–E; arrows indicate cells with HIV DNA and HIV-gp120 expression). However, as indicated in Figure 3E, cells lacking HIV DNA were also positive for the gp120 protein (Figure 3E, white staining. Higher amplification of the HIV-gp120 positive areas is presented in Appendix A). Additionally, it is important to note that several cells with integrated HIV DNA did not express HIV-gp120 (Figure 3 and Appendix A). The quantification of the total numbers of microglia/macrophages with integrated HIV DNA and expressing HIV-mRNA that were also positive for the HIV-gp120 protein indicated that a total of 1.98 ± 1.61 cells were triple positive within brain tissues obtained from HIV-infected individuals with undetectable replication (HIV_un_; Figure 3F). In brain tissues obtained from HIV-infected individuals with low and high systemic replication, triple-positive microglia/macrophages increased to 14.68 ± 7.1 and 189.4 ± 88.89 cells, respectively (Figure 3F; HIV_low_ and HIV_high_ conditions, * *p* ≤ 0.005 compared to uninfected conditions, # *p* ≤ 0.005 compared to HIV_un_ conditions, and & *p* ≤ 0.005 compared to HIV_un_ or HIV_low_ conditions, n = 34 tissues analyzed and 21 tissues compared to uninfected tissues, Un and Alz, n = 8 different tissues; each point represents 3–5 different areas per tissue analyzed). Thus, our data indicate that HIV-gp120 is still expressed in the ART era despite a significant reduction in the numbers of microglia/macrophages expressing this viral protein upon viral suppression (Figure 3F; F = 165,4816, by ANOVA to compare the different HIV groups).

In contrast, astrocytes triple-positive for HIV DNA, viral mRNA, and HIV-gp120 were stable and independent of the ART status (Figure 3F; astrocytes, HIV_un_, HIV_low_, or HIV_high_), suggesting significant differences compared to myeloid cells. The numbers of positive astrocytes in HIV undetectable replication conditions corresponded to 41.25 ± 12.49 astrocytes per tissue analyzed (Figure 3F; HIV_un,_ * *p* ≤ 0.005 compared to uninfected conditions; each point represents 3–5 different areas per tissue analyzed), HIV low-replication conditions corresponded to 37.42 ± 17.21 astrocytes per tissue analyzed (Figure 3F; HIV_low,_ * *p* ≤ 0.005 compared to uninfected conditions; each point represents 3–5 different areas per tissue analyzed), and HIV high-replication conditions corresponded to 37.25 ± 15.01 astrocytes per tissue analyzed (Figure 3F; HIV_high,_ * *p* ≤ 0.005 compared to uninfected conditions; each point represents 3–5 different areas per tissue analyzed. ANOVA analysis among the HIV-infected groups did not show any significant difference). Our data indicate that HIV reservoirs in microglia/macrophages are responsive to changes in systemic replication, but HIV-infected astrocytes are not. In contrast, in HIVE conditions, HIV-gp120 protein correlated with the total numbers of HIV-infected cells containing HIV DNA and HIV-mRNA and were mostly concentrated in macrophages than astrocytes (data not represented, 876 ± 458 cells for macrophages, and 624 ± 287 for astrocytes, triple-positive cells), again indicating that ART reduces the numbers of cluster and total numbers of microglia/macrophages, but not astrocytes, expressing HIV-gp120.

Similar to our HIV-p24 data, most HIV-gp120 staining was localized in cells lacking HIV DNA signal, but in close contact with HIV-infected cells (750.61 ± 164.85 cells, HIV DNA (−), Figure 3G, * *p* ≤ 0.005 compared to uninfected conditions and # *p* ≤ 0.005 compared to cells with HIV DNA (+)). In the brain samples obtained from individuals with undetectable, low, and high systemic HIV replication, the combined numbers of triple-positive cells corresponded to 10.45 ± 3.65 cells, suggesting that HIV-gp120 production, release, and bystander uptake were important mechanisms HIV-gp120 spread within the brain, even in the ART era (Figure 3G). In contrast, in HIVE brains, there was a perfect correlation between HIV-gp120 production in HIV DNA-positive cells (Figure 3G; HIVE DNA (+), 1345 ± 349, and HIVE DNA (−), 901 ± 302 cells).

Furthermore, in brain tissues obtained from individuals with undetected, low, and high HIV replication, we calculated the diffusion of the HIV-gp120 protein from the HIV-positive clusters reaching distances of up to 345.8 ± 89.71 µm (data not represented). All tissues showed a similar HIV-gp120 expression and distribution (Appendix A). In addition, the pixel intensity between long-term ART and HIVE conditions indicated that cells with high HIV-gp120 protein expression were reduced by 29.56 ± 8.71%. We did not find any correlation between HIV DNA, HIV-mRNA, HIV-gp120 protein expression, and cognitive impairment. No unspecific staining for HIV-gp120 and other viral components was detected in brain sections obtained from uninfected individuals (see Appendix A). Overall, our data indicate that HIV-gp120 protein is still synthesized, released, and taken up by neighboring uninfected cells despite long-term ART supporting multiple claims that residual gp120 production could be toxic.

**HIV-integrase is poorly expressed in the brain of HIV-infected individuals under effective ART.** Integrase is a key HIV enzyme required for HIV DNA integration into the host DNA [65,66,67]. Its association with neurotoxicity is unclear or controversial [68]. Our staining results indicate that integrase has a highly localized expression (Figure 4A–E) that does not correlate with the presence of clusters containing HIV-infected cells. However, as observed in the merge (Figure 4E), some cells had a nuclear accumulation of integrase (Figure 4; asterisks, * and **). Nonetheless, this viral protein’s overall expression was low compared to other HIV proteins, likely due to the nature and limited expression of integrase with the viral DNA [69]. The low integrase expression was not related to the antibody used because the integrase antibody and all other antibodies tested recognized integrase perfectly in OM-10 and ACH-2 cells, as indicated in the Methods Section and the Appendix A. The quantification of the integrase staining corroborates our observations that minimal to non-expression of integrase is present in the brains of individuals with undetectable, low, and high replication due to ART (Figure 4A–E; white arrows indicate cells with integrated HIV DNA. The asterisk indicates cells with nuclear integrase (Figure 4E)). The quantification of the numbers of microglia/macrophages containing HIV DNA, HIV-mRNA, and integrase protein showed that in brain tissues obtained from HIV-infected individuals with undetectable replication, the pool of cells expressing integrase corresponded to 0.75 ± 0.8 microglia/macrophages per tissue analyzed (HIV_un_; Figure 4F). The brain tissues obtained from HIV-infected individuals with low systemic replication corresponded to 6.53 ± 1.98 microglia/macrophages, suggesting that ART and systemic replication significantly impacted the numbers of cells expressing integrase (Figure 4F; HIV_un_ and HIV_low_). The differences in integrase expression within the brain and related to ART became evident when brains of individuals with high systemic replication were analyzed with 325.05 ± 85.69 cells positive for integrase in microglia/macrophages (Figure 4F; HIV_high_, * *p* ≤ 0.005 compared to uninfected conditions, # *p* ≤ 0.005 compared to HIV_un_ conditions, and & *p* ≤ 0.005 compared to HIV_un_ or HIV_low_ conditions; n = 34 tissues analyzed and 21 tissues compared to uninfected tissues, Un and Alz, n = 8 different tissues; each point represents 3–5 different areas per tissue analyzed. Comparisons of the HIV-infected groups for ANOVA indicated that HIV_high_ was significantly different than HIV_un_ and HIV_low_ groups; F = 552.37). Furthermore, the pixel intensity for integrase corresponded only to 8.54 ± 3.75% of the pixel intensity observed for HIV-gp120 and HIV-p24 in HIV_un_ and HIV_low_ conditions in microglia/macrophages (data not represented).

In contrast, in HIV-infected astrocytes producing viral mRNA, integrase was expressed in higher amounts than HIV_un_ and HIV_low_ in microglia/macrophages (Figure 4F). However, there were no differences in the numbers of triple-positive cells, despite the differences in systemic replication, 26.38 ± 6.5 cells (HIV_un_), 20.85 ± 4.27 cells (HIV_low_), and 25.35 ± 9.7 cells (HIV_high_), suggesting an even lower expression or stability of integrase in HIV-infected astrocytes (Figure 4F; astrocytes). If we compare these data to HIVE conditions, integrase was expressed in 1136 ± 378 cells also positive for HIV DNA and viral mRNA (HIVE; data not shown) corresponding to 905 ± 357 microglia/macrophages and 569 ± 299 astrocytes (HIVE; data not shown), supporting the idea that ART or better disease management reduces the expression of integrase in microglia/macrophages and astrocytes.

To quantify whether integrase was accumulated in uninfected cells, as described for HIV-p24 and HIV-gp120, we quantified colocalization with HIV DNA. Our data indicate that integrase is expressed at low levels in HIV DNA (+) cells, 11.22 ± 8.52 cells (Figure 4G and compared to Figure 4F; * *p* ≤ 0.005 compared to uninfected conditions). The quantification of integrase in HIV DNA-negative cells highlighted that only 4.21 ± 2.46 cells were positive, suggesting that integrase was poorly secreted and accumulated viral protein in neighboring cells (Figure 4G; HIV DNA (−). * *p* ≤ 0.05 compared to uninfected conditions). In contrast, the examination of HIVE tissues indicated that most HIV DNA (+) cells, 1406 ± 425 cells (Figure 4G; HIVE DNA (+)) were positive for integrase and HIV DNA-negative cells corresponded to 905 ± 357 cells (Figure 4G; HIVE DNA (−). * *p* ≤ 0.005 compared to uninfected conditions). Unspecific staining was not observed in tissues obtained from uninfected individuals (see Appendix A). Thus, the expression of integrase protein was low but significant compared to uninfected tissues but was not comparable to the expression or distribution observed for other HIV proteins, such as HIV-p24 and gp120.

**HIV-nef protein is expressed in HIV-infected cells and is taken up by uninfected cells, even in individuals without detectable systemic replication**. HIV-nef is involved in many neuronal/glial/immune toxicity mechanisms, even in the current ART era [70,71,72]. Thus, we evaluated HIV-nef expression and distribution, as described for the other HIV proteins.

Experiments using brains from HIV-infected individuals with undetectable and low systemic viral replication indicated that HIV-nef was expressed in association with HIV DNA-positive clusters. Here, we showed a selected region containing HIV DNA-positive cells containing the HIV-nef protein; however, we must note that most brain areas were negative for HIV DNA and HIV-nef protein (Figure 5A–E). Confocal analysis identified cells containing HIV DNA (Figure 5A–E; arrows), but also cells with high nef accumulation but negative for HIV DNA (Figure 5A–E; see asterisks, * and **, denote cells with high expression or accumulation (Figure 5E)).

The quantification of HIV-nef protein expression in microglia/macrophages indicated that triple-positive cells (HIV DNA/viral mRNA and HIV-nef protein) were present in all HIV conditions, including undetected, low, and high systemic replication (Figure 5F; microglia/macrophages) with an abundance of 10.65 ± 4 cells (HIV_un_), 40.4 ± 20.50 (HIV_low_), and 371.48 ± 70.82 cells (HIV_high_) per tissue analyzed (Figure 5F). Interestingly, a similar trend was observed for the other viral proteins analyzed, HIV-p24, HIV-gp120, and integrase, in correlation with systemic viral replication (Figure 5F; * *p* ≤ 0.005 compared to uninfected conditions, # *p* ≤ 0.005 compared to HIV_un_ conditions, and & *p* ≤ 0.005 compared to HIV_un_ or HIV_low_ conditions; n = 34 tissues analyzed and 21 tissues compared to uninfected tissues, Un and Alz, n = 8 different tissues; each point represents 3–5 different areas per tissue analyzed. Comparisons of the HIV-infected groups for ANOVA indicate that HIV_high_ was significantly different than HIV_un_ and HIV_low_ groups; F = 883.03).

In astrocytes, HIV-nef protein expression in HIV DNA and HIV-mRNA-positive astrocytes was significant in all HIV conditions, including undetectable, low, and high systemic replication (Figure 5F; astrocytes). The numbers of positive astrocytes were 25.28 ± 15.62 cells and 43.73 ± 27.21 in undetectable (HIV_un_) and low replication (HIV_low_) (Figure 5F; * *p* ≤ 0.005 compared to uninfected conditions, # *p* ≤ 0.005 compared to HIV_un_ conditions, and & *p* ≤ 0.005 compared to HIV_un_ or HIV_low_ conditions; n = 34 tissues analyzed and 21 tissues compared to uninfected tissues, Un and Alz, n = 8 different tissues; each point represents 3–5 different areas per tissue analyzed). In brain tissues obtained from individuals with high systemic viral replication, astrocytes corresponded to 247.88 ± 118.13 cells per tissue analyzed. However, and in contrast to the analyses of HIV-p24, HIV-gp120, and integrase, HIV-nef protein expression in astrocytes were dependent on systemic viral replication (Figure 5F; astrocytes. ANOVA indicates that HIV_high_ was significantly different than HIV_un_ and HIV_low_ groups; F = 122.59) in a similar manner to other viral proteins in microglia/macrophages. Additionally, the quantification of HIV-nef protein expression in HIVE tissues indicated that the total number of cells with HIV DNA, HIV-mRNA, and HIV-nef protein corresponded to 1233 ± 277 (HIVE; data not represented). In HIVE tissues, most of these triple-positive cells were microglia/macrophages, 975 ± 345, and a small population was astrocytes, 605 ± 278. Overall, the data indicate that HIV-nef protein expression in microglia/macrophages and astrocytes depends on systemic viral replication. These data indicate that long-term ART can affect different viral proteins in a cell type-specific manner.

To quantify whether HIV-nef protein could be released into neighboring uninfected cells, we analyzed HIV DNA-positive and -negative cells, as described above. The quantification of HIV-nef protein in HIV DNA and HIV-mRNA indicated that 17.15 ± 2.81 cells per tissue were analyzed (Figure 5G; HIV DNA (+)). However, most HIV-nef localized in HIV DNA-negative cells, 341.63 ± 107.68 cells, but was always in close contact with clusters of HIV DNA-positive cells, suggesting that most HIV-nef protein accumulated in the surrounding uninfected cells (Figure 5G; HIV DNA (−). * *p* ≤ 0.005 compared to uninfected conditions, n = 21 tissues compared to uninfected tissues, n = 8; each point represents 3–5 different areas per tissue analyzed). The greater magnification of the HIV-nef distribution in brains obtained from individuals with undetectable replication denoted the specificity and wide distribution of HIV-nef (Appendix A; arrows denote cells with integrated DNA). In contrast, in HIVE conditions, most HIV-nef proteins were associated with HIV DNA-positive cells, 1468 ± 236 cells, and only 958 ± 387 cells without integrated HIV DNA showed nef protein (HIVE DNA (−), Figure 5G, * *p* ≤ 0.005 compared to uninfected conditions). No unspecific staining for HIV DNA or HIV-nef was observed in tissues obtained from uninfected individuals (see Appendix A). The quantification of the pixel intensity between HIV_un_ and HIV_low_ (see Table 1) and HIV_high_ and HIVE conditions indicated that cells with high HIV-nef protein intracellular content only had a reduction in HIV-nef protein staining of 33.78 ± 7.09%, compared to HIV_high_ and HIVE conditions, suggesting that specific areas of the brain can accumulate high amounts of HIV-nef protein, even in the ART era. Additionally, we calculated that HIV-nef diffused from HIV DNA-positive cells into neighboring uninfected cells reaching distances of 493.02 ± 104.5 µm from the clusters of HIV-infected cells, suggesting an active mechanism of synthesis, release, and bystander uptake (data not shown). Overall, our data indicate that HIV-nef protein is still synthesized, released, and taken up by neighboring uninfected cells, despite the suppression of systemic viral replication.

**HIV-vpr protein is expressed within the brain of HIV-infected individuals under ART.** HIV-vpr is the multifunctional HIV accessory protein essential for transporting the pre-integration complex into the nucleus [73]. HIV-vpr has also been associated with the pathogenesis of NeuroHIV, but whether this viral protein is expressed within the human brain in the ART era is unknown.

To evaluate the expression and distribution of HIV-vpr, the staining of brain tissues obtained from uninfected and HIV-infected individuals with undetectable (HIV_un_), low (HIV_low_, and high (HIV_high_) replication for HIV-vpr protein and DAPI, HIV DNA, and HIV mRNA was performed (Figure 6A–E). The staining analysis indicated a low expression and scattered distribution (Figure 6A–E; the arrows denote cells with integrated HIV DNA and asterisks denote cells with high HIV-vpr accumulation). In the case of HIV-vpr, the protein could not establish a diffusion pattern from HIV-infected clusters as established for HIV-gp120 and HIV-nef; however, HIV-vpr, similar to HIV-nef, also highly accumulated in particular cells (Figure 6E; asterisks). The quantification of the numbers of microglia/macrophages per tissue analyzed indicated that HIV-vpr expression depended on systemic viral replication with 1.88 ± 1.67 cells in tissues obtained from individuals with undetectable replication (HIV_un_), 34.08 ± 21.5 cells in low-replication, and 222.5 ± 134.59 cells in high-replication conditions (Figure 6F, microglia/macrophages). Interestingly, there was a correlation between systemic replication and the numbers of HIV-infected microglia/macrophages (Figure 6F; * *p* ≤ 0.005 compared to uninfected conditions, # *p* ≤ 0.005 compared to HIV_un_ conditions, and & *p* ≤ 0.005 compared to HIV_un_ or HIV_low_ conditions; n = 34 tissues compared to uninfected tissues; each point represents 3–5 different areas per tissue analyzed).

The quantification of astrocytes containing HIV DNA, HIV-mRNA, and HIV-vpr indicated that HIV-vpr protein expression depended on systemic viral replication (Figure 6F; astrocytes). The astrocyte numbers corresponded to 35.55 ± 13.07 in brain tissues obtained from individuals with undetectable systemic replication (HIV_un_), 59.5 ± 18.83 in brain tissues obtained from individuals with low systemic replication (HIV_low_), and 74.63 ± 21 cells in brain tissues obtained from individuals with high systemic replication (HIV_high_) (Figure 6F; astrocytes. * *p* ≤ 0.005 compared to uninfected conditions, # *p* ≤ 0.005 compared to HIV_un_ conditions, and & *p* ≤ 0.005 compared to HIV_un_ or HIV_low_ conditions; n = 34 tissues analyzed, and 21 tissues compared to uninfected tissues, Un and Alz, n = 8 different tissues; each point represents 3–5 different areas per tissue analyzed). In HIVE conditions, 1198 ± 315 of the cells were positive for HIV DNA, HIV-mRNA, and HIV-vpr. Most of the cells were microglia/macrophages, 846 ± 208 of the cells, and a small population of astrocytes, 623 ± 277 cells (HIVE; data not represented). In addition, the pixel intensity was 74.08 ± 18.39% lower in HIV_un/low_ than in HIV_high_ and HIVE conditions (data not represented).

To evaluate the bystander uptake of HIV-vpr, we spatially quantified HIV-vpr in HIV DNA-positive and -negative cells. In a similar manner to HIV-gp120 and HIV-nef data, most infected cells expressed residual vpr (Figure 6G; 9.68 ± 2.70 cells); however, most HIV-vpr signals were located in uninfected cells (Figure 6F; HIV DNA-negative cells, HIV DNA (−), and 855.83 ± 257.45), but we could not establish an association or diffusion patterns similar to HIV-gp120 and HIV-nef proteins (Figure 6G; * *p* ≤ 0.005 compared to uninfected conditions and # *p* ≤ 0.005 compared to HIV DNA (+) cells). In HIVE conditions, most HIV-vpr was located in HIV-positive DNA cells, 1289 ± 245 cells, and 920 ± 105 cells with HIV-negative signals (Figure 6G; HIVE DNA (+) or (−)). Unspecific staining for HIV DNA or HIV-vpr was not observed in tissues obtained from uninfected individuals (see Appendix A). Overall, HIV-vpr synthesis and bystander uptake are still significant in the current ART era.

**HIV-tat protein is expressed and secreted within the CNS.** HIV-tat corresponds to the transactivator of the virus, and its main function is to drive HIV transcription [74,75]. Multiple laboratories, including ours, identified that soluble HIV-tat has neurotoxic effects and has even been reported in the CSF of HIV-infected individuals under ART [76]. However, it is still debatable whether HIV-tat is produced significantly during long-term HIV infection and the current ART era.

The staining of brain tissues obtained from uninfected and HIV-infected individuals with undetectable, low, and high systemic replication for HIV-tat protein, DAPI, HIV DNA, and HIV-mRNA shows that HIV-tat was expressed and widely distributed around clusters of cells containing HIV DNA (Figure 7A–E; arrows represent cells with integrated HIV DNA signals and the asterisk represents cells with high HIV-tat accumulation). Additionally, as described for HIV-gp120, HIV-nef, and HIV-tat in this paper, the viral protein distribution was highly localized in brain areas with clusters of HIV-infected cells; however, most of the tissue was negative for viral components (Figure 7A–E; representing a cluster of HIV-infected cells). We calculated that HIV-tat could diffuse from HIV DNA-positive clusters into neighboring uninfected cells, reaching distances of up to 492.1 ± 145.68 µm. The quantification of the total numbers of microglia/macrophages with integrated DNA, producing HIV-mRNA, and positive for HIV-tat showed that, in the brain obtained from individuals with undetectable replication, the numbers of triple-positive cells corresponded to 8.38 ± 4.4 cells (HIV_un_; Figure 7F), in samples with low replication the numbers were 48.8 ± 15.9 cells (HIV_low_; Figure 7F), and in samples with high replication the numbers corresponded to 365.68 ± 68.05 cells (HIV_high_; Figure 7F. microglia/macrophage, * *p* ≤ 0.005 compared to uninfected conditions, # *p* ≤ 0.005 compared to HIV_un_ conditions, and & *p* ≤ 0.005 compared to HIV_un_ or HIV_low_ conditions; n = 34 tissues analyzed and 21 tissues compared to uninfected tissues, Un and Alz, n = 8 different tissues, each point represents 3–5 different areas per tissue analyzed). Interestingly, the numbers of triple-positive cells were inversely correlated with the degree of systemic replication, suggesting that systemic replication had a significant impact on the production of HIV-tat within the brain (Figure 7F). However, HIV-tat production is still highly significant, even in HIV-infected individuals with undetectable replication. The ANOVA analysis of the microglia/macrophages HIV-infected groups indicated that brain tissues from low and high viral systemic replication were significantly different compared to brains obtained from individuals with undetectable replication (Figure 7F; comparisons of the HIV-infected groups for ANOVA indicate that HIV_high_ is significantly different compared to HIV_un_ and HIV_low_ groups (F = 936.96)).

The HIV-tat accumulation in astrocytes indicated that HIV-tat expression was systemic viral-replication-dependent (Figure 7F; astrocytes). The number of triple-positive astrocytes in tissues obtained from individuals with undetectable replication was 50.53 ± 18.03 cells (HIV_un_, Figure 7F, astrocytes), with low replication was 143.93 ± 102.1 cells, and with high replication was 385.25 ± 66.34 cells (Figure 7F; astrocytes. * *p* ≤ 0.005 compared to uninfected conditions, # *p* ≤ 0.005 compared to HIV_un_ conditions, and & *p* ≤ 0.005 compared to HIV_un_ or HIV_low_ conditions; n = 34 tissues analyzed, and 21 tissues compared to uninfected tissues, Un and Alz, n = 8 different tissues; each point represents 3–5 different areas per tissue analyzed). A high-magnification representative reconstruction denoted the HIV-tat distribution and concentration of the protein in specific cell types (Appendix A; arrows represent the cells with HIV DNA). In astrocytes, HIV-tat expression depended on systemic viral replication, such as HIV-nef and HIV-vpr, but not HIV-p24 and integrase, suggesting a different expression mechanism and ART-mediated regulation. The accumulation of tat in astrocytes followed a similar distribution to nef and vpr, but not for HIV-p24 or integrase, suggesting differential stability or expression for some viral proteins even in the current ART era. In contrast, in HIVE conditions, a total of 1193 ± 244 cells were identified in a similar volume analyzed in samples under long-term ART. Most HIV-infected cells containing HIV-tat proteins were microglia/macrophages, 801 ± 197, and a small population of astrocytes, 685 ± 297 (data not represented).

To evaluate the bystander release and uptake, we quantified the amount of HIV-tat in cells with HIV DNA and neighboring uninfected cells (Figure 7G). HIV-tat was present in most HIV-infected cells (HIV DNA ((+), Figure 7G, 13.96 ± 3.65 positive cells for HIV DNA and HIV-tat), but most HIV-tat proteins were accumulated in neighboring uninfected cells lacking HIV DNA or gag-pol mRNA, 1383.71 ± 164.85 (HIV DNA (−); Figure 7G). The quantification of HIV-tat-positive cells’ pixel intensity indicated two populations of cells: first, a low but consistent HIV-tat intracellular expression (cytoplasm and nucleus) localization and a second profile with high cytoplasmic/nuclear accumulation. Interestingly, the expression of HIV-tat in neighboring cells could reach numbers similar to HIVE (Figure 7G; HIVE in HIV-positive and -negative cells), indicating that, even in the ART era, some areas of the brain still produce and accumulate significant amounts of HIV-tat. In HIVE conditions, most HIV-tat was localized in microglia/macrophages, 1306 ± 306 cells, and 934 ± 277 cells corresponded to astrocytes (Figure 7G; * *p* ≤ 0.005 compared to uninfected conditions and # *p* ≤ 0.005 compared to HIV DNA ((+)) cells). No unspecific staining for HIV-tat was observed in tissues obtained from uninfected individuals (see Appendix A). Overall, we demonstrated that HIV-gp120, HIV-nef, HIV-vpr, and HIV-tat proteins, but not integrase, were still produced in a significant manner within the brains of HIV-infected individuals, but the expression and distribution were dependent on systemic viral replication.

## 5. Discussion

The current study presents a method to detect HIV DNA, viral mRNA, and proteins in a cell-type-dependent manner and with a high spatial resolution in brain tissues, even in tissues obtained from HIV-infected individuals under long-term ART. Our findings include: first, a novel method to detect and quantify viral reservoirs (silent, active, and residual mRNA and protein production) within the CNS of HIV-infected individuals; second, we provided the distribution of viral reservoirs within the brain; third, we provided the quantity and cell type with integrated HIV DNA; fourth, we provided the quantity and cell type with integrated HIV DNA that still produces viral mRNA and proteins within the CNS of individuals under ART; fifth, we identified that several viral proteins, but not all, are produced, secreted, and taken up by neighboring uninfected cells; sixth, we identified that efficient systemic ART reduces the brain reservoir pool and prevents the synthesis of some viral proteins, but not all of them; and, lastly, our data provided the size of the viral reservoirs within the CNS, and the foundation to support the fact that those viral proteins are still secreted in the current ART era.

A key feature of HIV infection that has made it virtually impossible to cure this disease is the early generation of latent viral reservoirs in different tissues, including the central nervous system (CNS). By definition, a viral reservoir corresponds to long-living infected cells, mainly localized in a specific anatomical compartment, where the replication-competent virus can persist for a longer period of time than the main pool of actively replicating viruses [3,5,77,78]. The best-known viral reservoir corresponds to a small pool of CD4^+^ T cells in the circulation and tissue sanctuaries with a low drug penetrance or biological barrier that enables survival and residual replication [79,80,81]. Several tissue-specific viral reservoirs have been proposed, including macrophages in multiple tissues, dendritic cells, astrocytes, and pericytes [29,82]. However, whether these cells are viral reservoirs in vivo and their size in the current ART era are still debated and deserve further investigation. A major obstacle to answering the previously mentioned concerns is the lack of techniques available to detect tissue viral reservoirs [83]. To date, with ultrasensitive methods (blood-based systems), we still notice residual viremia in patients under ART, suggesting that residual viral replication still occurs in tissues [1,84,85]. The present study focused on the characterization, location, and size of these viral populations in the current ART era.

The brain has been proposed to be a major viral reservoir due to its immune privilege and the blood-brain barrier that limits immune trafficking/surveillance and some ART drug penetration, suggesting a different evolution of the reservoirs and the virus itself. In agreement, HIV sequences are diverse upon viral rebound and not only from circulating CD4^+^ T cells, suggesting that rebounded viruses come from different reservoirs [1]. Furthermore, phylogenetic studies indicate that HIV production from the brain is associated with viral resistance to treatment and HIV-associated neurocognitive disease (HAND) [86,87]. Thus, a new understanding of the brain’s role is emerging only recently, especially because at least 50% of the HIV-infected population has HAND signs even in the current ART era. This suggests that chronic inflammation still occurs in the brain and is independent of systemic replication. HAND’s pathogenesis is currently evolving from the early stages of AIDS, in 1980–1990, to the introduction of ART, which transforms HIV disease into a chronic condition. The cognitive impairment has been associated with ART’s neurotoxicity, chronic inflammation, comorbidities, the presence of quiescent/latent infected cells, and the residual production and secretion of viral proteins. However, most of these mechanisms have been examined using in vitro and several animal models, but whether or not these mechanisms are present in vivo, the extent of the associated damage or the size of the viral reservoirs, and the viral protein synthesis are poorly explored in human tissues. We were surprised to find that ART and changes in systemic replication reduced the size of the reservoirs in the brain, but residual viral protein synthesis and secretion were differentially affected by ART in a cell-type-dependent manner. These ART-mediated changes in viral protein synthesis need to be addressed in future studies to understand them.

Several groups provided the rules to classify a viral reservoir: first, the presence of integrated HIV DNA in long-lasting cells; second, the mechanism of extended survival to perpetuate the virus; and third, the mechanism of viral silencing, reactivation, and transfer into cells that support high replication. Our data indicate that both microglia/macrophages (Iba-1-positive cells) and a small population of astrocytes (GFAP-positive cells) contain integrated HIV DNA in the host DNA. Both populations remain stable within the CNS for years, probably due to the multiple mechanisms of survival triggers for HIV latency [28,30,88,89,90]. We demonstrated that both cell types could transfer infection into cells that support high replication, as described [31,33]. Our endeavor to quantify the cell type and frequency of these HIV-infected cells in large tissue areas provide a more accurate description of viral reservoirs within the brain in the current ART era due to several factors. The imaging system’s specificity developed cell-type identification, 3D reconstructions, and the evolution of several viral markers (HIV DNA, HIV-mRNA, and viral proteins). Our data identify viral reservoirs, classify them into particular cell types, and provide critical information on residual viral protein expression and secretion into neighboring uninfected cells. This bystander viral protein mechanism supports multiple in vitro data about the toxic role of viral protein, even in the current ART era. In addition, we identified that long-term ART reduced the pool of myeloid viral reservoirs but minimally reduced the pool of HIV-infected astrocytes. Furthermore, ART differentially affected viral protein expression in microglia/macrophages versus astrocytes, suggesting different translation mechanisms or stability.

We recently identified that latently HIV-infected microglia/macrophages and astrocytes are protected from apoptosis by several mechanisms, including preventing apoptosome formation and changing their metabolism; importantly, using cell-to-cell communication mechanisms to spread toxicity and local inflammation [31,32,51,89,91,92]. Local events in the current ART era are a critical point used to denote the presence of viral reservoirs and associated mechanisms of survival and inflammation. Latently HIV-infected microglia and macrophages expressed high quantities of the Bim protein, preventing pore formation and subsequent apoptosis [30]. We detected, in latently infected cells, microglia/macrophages and astrocytes, that mitochondria were highly affected, even in the absence of viral replication due to interorganelle compromise [91]. Furthermore, we identified that CNS viral reservoirs relied on an unusual source of carbon, such as glutamine/glutamate, an abundant source of energy within the brain, instead of glucose and fatty acids, to survive and remain for extended periods in the brain [93]. Blocking the use of glutamine/glutamate resulted in the apoptosis of latently infected cells because cells with integrated HIV DNA could not change carbon sources to produce energy [91]. The behavior of HIV-infected astrocytes is puzzling compared to microglia/macrophages due to long-term infections. ART minimally reduced the pool of HIV-infected cells within the brain. Long-term ART did not prevent the synthesis and release of particular viral proteins, suggesting that the mechanisms of viral silencing and apoptosis of microglia/macrophages and astrocytes are different. Only recently, using primary human astrocytes and long-term infection protocols, we identified that human astrocytes infected with HIV were protected from apoptosis by preventing apoptosome formation by compromising interorganelle interactions [32]. Furthermore, we identified that astrocytes could prevent silent viral replication, even in the absence of ART. Additionally, methamphetamine, SAHA, and cytokines can reactivate the virus and transfer it into immune cells, but the virus becomes silent again, suggesting an intrinsic mechanism of silencing, reactivation, and efficient transfer into immune cells not observed in other viral reservoirs [31]. We propose that these mechanisms explain the predominance of astrocytes as a glial reservoir over microglia/macrophages in long-term HIV infection and ART use.

Furthermore, we identified that viral reservoirs in vivo and in vitro used cell-to-cell communication systems to spread local toxicity, inflammation, and infection by both tunneling nanotube (TNT) and gap junction-dependent mechanisms [94,95]. Both mechanisms underscored the importance of isolating viral reservoirs in the current ART era (long distances between them) and these cells’ capacity to communicate with uninfected surrounding cells and immune cells to spread toxicity and viral infection. Our recent data indicate that TNTs are essential in the early stages of infection, as well as during reactivation by the transport of viral components at long distances in tissues, up to 500 µm, supporting the exceptional mechanism of cell-to-cell communication present in viral reservoirs to survive detection and spread local inflammation [94,95]. These data indicate that viral reactivation is not required to generate damage.

Astrocytes are the most abundant cells within the CNS, with up to 20% of the cells with integrated HIV DNA in the early years of ART; however, recently obtained data indicate the lack of HIV DNA, further contributing to the discussion of whether or not astrocytes are a viral reservoir [37,96]. Our data indicate that most residual viral protein expression in the current ART era comes from these cells. A critical point that needs to be discussed further is the localization of viral reservoirs and the formation of clusters between microglia/macrophages and the surrounding astrocytes, complicating the interpretation of viral reservoir identification. Our data reveal that clusters with integrated HIV DNA contain HIV DNA-positive astrocytes. Our laboratory recently demonstrated that viral reservoirs could amplify local inflammation into neighboring uninfected cells by maintaining gap-junctional and hemichannel communication by an HIV-tat-mediated mechanism [97,98]. Our data, especially concerning brains obtained from HIV-infected individuals with undetectable or low replication, strongly indicate that viral proteins, such as HIV-gp120, HIV-nef, and HIV-tat, are still produced and could explain the long-term toxicity of HIV in the brain. We must denote that residual viral protein secretion by some viral reservoirs (myeloid and glial) could expand into neighboring uninfected cells, including neurons and other cell types, suggesting specific mechanisms of uptake and toxicity not addressed in the current study, but all viral proteins require further investigation to obtain a cure. For example, some viral proteins, such as HIV-tat, are taken up for neurons by the low-density lipoprotein receptor-related protein (LRP1). In this case, as well as in endothelial cells, a specific mechanism of toxicity amplification is mediated by nitric oxide and the dysregulation of NMDA receptors or by endothelial barrier dysfunction [99,100,101]. These mechanisms of viral-mediated inflammation are novel and probably cell-specific, requiring further investigation.

One of the consequences of HIV infection in the brain is neuronal cell death and CNS inflammation due to immune activation, oxidative stress, and neurotoxicity. It has been suggested that HIV infects astrocytes and microglia/macrophages in the CNS, but not neurons. Although neurotoxic substances, such as HIV proteins (HIV-tat, HIV-gp120, and HIV-nef) are released by infected astrocytes and microglia/macrophages, the mechanisms of synaptic compromise and toxicity in neurons are still under active investigation [102,103]. It has been determined that the transactivator of the transcription protein, tat, can bind to the surface of the neurons and increase the intracellular level of calcium, synapsis loss, and neuroinflammation [97,104,105,106,107,108]. The HIV envelope glycoprotein, HIV-gp120, has a neurotoxic effect on dopaminergic neurons, impairs the structure and function of the neurons’ cytoskeleton, and causes mitochondrial damage in neurons [109,110,111,112,113,114]. HIV-nef is known to be toxic to neurons, but the neurotoxic effect of HIV-nef is not well characterized [115]. Although, it has been reported that HIV-nef activates astrocytes and microglia, releasing pro-inflammatory cytokines [116,117]. Our results show that, in large brain areas of patients in long-lasting ART, astrocytes and microglia/macrophages are viral reservoirs that still produce viral proteins. These viral reservoirs can be continuously reactivated, promoting residual viral replication and the production of selected viral proteins that promote bystander neurotoxicity. We must note that viral protein accumulation depends on the distance from the viral reservoir and the cell type analyzed. These findings support multiple claims of bystander damage and cognitive impairment associated with HIV infection, even in the current ART era. They also corroborate the idea that eradicating the viral reservoir in the CNS is a major barrier to improving the treatment and recovery of HIV-infected patients’ neurological functions.

Nevertheless, several questions remain relevant in the field, including the contribution of RNA-splicing species to viral reservoir residual replication and establishing a clear correlation between bystander secretion/uptake of viral proteins and cognitive impairment. Our study still questions whether the detected proteins are part of a virion or correspond to disorganized complexes of viral proteins. Also, whether comorbidities such as drug abuse change the size of the viral reservoir pool or alter bystander damage within the CNS. All these points can be addressed with the current technology and hard work to identify the specific mechanisms of viral reservoir survival, bystander damage, and contribution of comorbidities to prevent or treat NeuroHIV and establish a cure.

Overall, the quantification of viral reservoirs in large areas of the human brain underscores several key points; first, the brain needs to be considered for cure strategies; second, viral reservoirs within the CNS are present, even in the current ART era; third, viral reservoirs in the brain are comprised of microglia/macrophages as well as astrocytes; fourth, residual viral replication still occurs; and fifth, bystander protein uptake still occurs in the current ART era.

In conclusion, viral reservoirs in the human brain have significant residual replication that could explain the cognitive disease observed in at least half of the HIV-infected population. Our data provide the foundation for future studies to understand the mechanism of viral reservoir localization, survival, and viral replication within the CNS. The elimination of these brain-associated viral reservoirs in long-lasting cells, microglia/macrophages, and astrocytes needs to be considered for cure and eradication efforts due to the significant differences in the anatomical sites and their differences to CD4^+^ T cells.

## 6. Conclusions

HIV-infected microglia/macrophages and a small population of astrocytes are the main brain cell types infected in the pre- and current ART era.HIV brain reservoirs still produce residual viral mRNAs and protein expression despite ART.HIV-p24, gp120, nef, vpr, and tat proteins, but not integrase, are expressed in HIV-infected cells, released, and taken up by neighboring uninfected cells, even in ART conditions.

## Figures and Tables

**Figure 1 cells-11-02379-f001:**
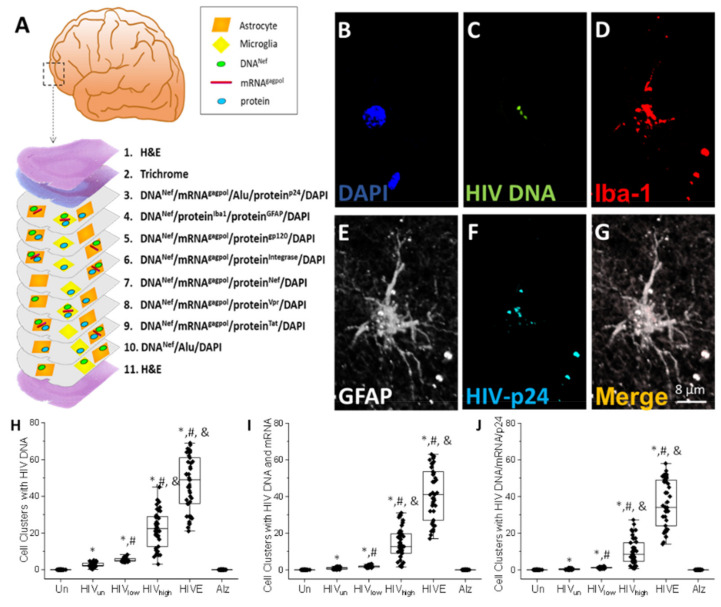
**HIV reservoirs are organized in small clusters within the brain of HIV-infected individuals.** (**A**) The image shows the detection strategy for HIV DNA, viral mRNA, and viral proteins, as well as cellular markers using 11 slice sections per tissue. The first and last sections were stained for H&E; the second section was used for a trichrome stain; the following sections were sequentially stained for DAPI, HIV DNA, HIV-mRNA, HIV protein (p24, gp120, integrase, nef, vpr, or tat), Alu repeats, or markers for astrocytes (GFAP), and microglia/macrophages (Iba-1) as indicated. (**B**–**G**) Confocal images show a cell cluster labeled with DAPI, containing HIV DNA, positive for microglia/macrophage markers and enlarged astrocytes, indicated by the Iba-1 protein and GFAP, respectively, and expressing HIV-p24 protein cells. Most brain tissue was negative for the viral markers (compared to Appendix A). (**H**–**J**) Quantification of cell clusters positive for HIV DNA in tissues obtained from uninfected brains (Un), HIV-infected brains with undetectable replication (HIV_un_), HIV-infected brains with low replication (HIV_low_), and HIV-infected brains with high replication (HIV_high_). As a control, we used HIV encephalitic brains (HIVEs) and uninfected cases from healthy and Alzheimer’s individuals, Alz (n = 34 brains analyzed and each point corresponding to the mean of at least 3–5 different quantifications per tissue). (**H**) Corresponds to the quantification of cell clusters containing HIV DNA in the nucleus (colocalizing with DAPI and Alu repeats). (**I**) Quantification of double-positive cell clusters for HIV DNA and viral mRNA. (**J**) Quantification of cell clusters positive for HIV DNA, viral mRNA, and HIV-p24. * *p* ≤ 0.005 compared to uninfected conditions, # *p* ≤ 0.005 compared to HIV_un_ conditions, and & *p* ≤ 0.005 compared to HIV_un_ or HIV_low_ conditions; n = 34 tissues analyzed, and 21 tissues compared to uninfected tissues, Un and Alz, n = 8 different tissues; each point represents 3–5 different areas per tissue analyzed. Bar: 8 µm.

**Figure 2 cells-11-02379-f002:**
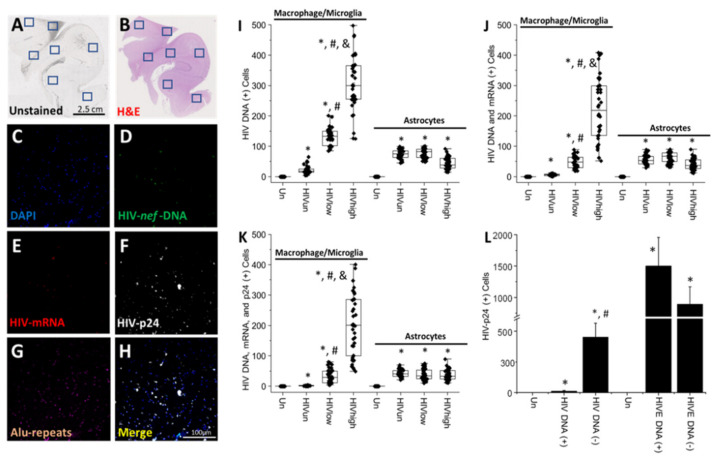
**Quantification of myeloid and glial viral reservoirs in cortical and subcortical human brain areas.** Representative confocal images of the areas analyzed for (**A**) unstained and (**B**) H&E-stained tissue. Areas detected with HIV DNA are represented in blue boxes. (**C**–**H**) Representative confocal images of the brain areas containing integrated HIV DNA and positive labeling for (**C**) DAPI, (**D**) HIV DNA, (**E**) HIV-mRNA, (**F**) HIV-p24, (**G**) Alu repeats, and (**H**) the merge of all colors. (**I**–**K**) Quantification of cells positive for (**I**) integrated HIV DNA in Macrophage/Microglia (Iba-1-positive cells) and astrocytes (GFAP-positive cells) in human tissues obtained from individuals with undetectable, low, and high systemic replication; * *p* ≤ 0.005 compared to uninfected conditions, # *p* ≤ 0.005 compared to HIV_un_ conditions, and & *p* ≤ 0.005 compared to HIV_un_ or HIV_low_ conditions; n = 34 tissues analyzed and 21 tissues compared to uninfected tissues, Un and Alz, n = 8 different tissues; each point represents 3–5 different areas per tissue analyzed. Comparisons of the HIV-infected groups by ANOVA are described in the text and Methods Section, (**J**) integrated HIV DNA and HIV-mRNA in Macrophage/Microglia and astrocytes in human tissues obtained from individuals with undetectable, low, and high systemic replication (* *p* ≤ 0.005 compared to uninfected conditions and # *p* ≤ 0.005 compared to HIV conditions, & *p* ≤ 0.005), (**K**) integrated HIV DNA, HIV-mRNA, and HIV-p24 in Macrophage/Microglia and astrocytes in human tissues obtained from individuals with undetectable, low, and high systemic replication (* *p* ≤ 0.005 compared to uninfected conditions, # *p* ≤ 0.005 compared to HIV_un_ conditions, and & *p* ≤ 0.005 compared to HIV_un_ or HIV_low_ conditions; n = 34 tissues analyzed and 21 tissues compared to uninfected tissues, Un and Alz, n = 8 different tissues; each point represents 3–5 different areas per tissue analyzed). (**L**) Quantification of cells expressing HIV-p24 in HIV and HIVE cells positive for HIV DNA (HIV DNA (+) and HIVE-DNA (+)) and negative for HIV DNA (HIV DNA (−) and HIVE-DNA (−)) (* *p* ≤ 0.005 compared to uninfected conditions and # *p* ≤ 0.005 compared to HIV DNA (+) conditions). Bar: 100 µm.

**Figure 3 cells-11-02379-f003:**
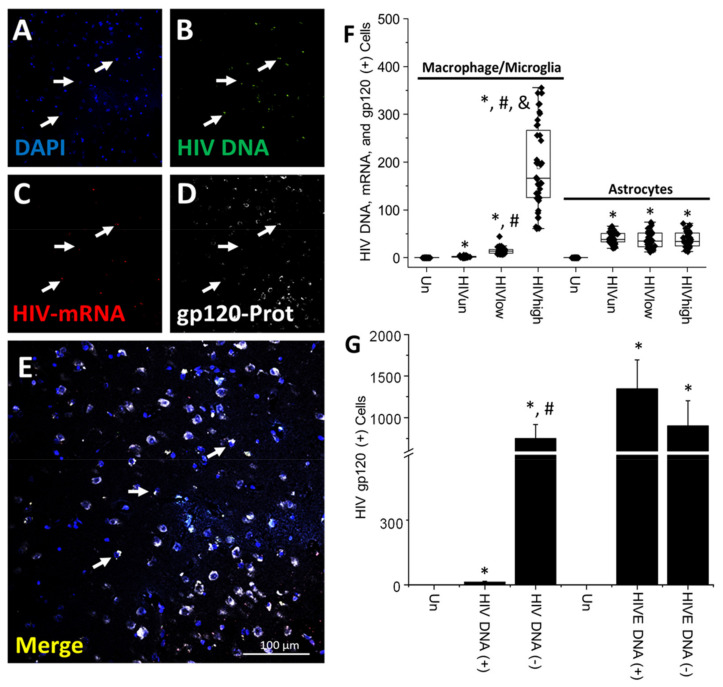
HIV-gp120 protein is expressed in myeloid and glial viral reservoirs and surrounding uninfected cells: potential bystander toxicity under the current ART era. (**A**–**E**) Representative confocal images showing (**A**) DAPI staining, (**B**) HIV DNA, (**C**) Macrophage/Microglia marker Iba-1 protein, (**D**) HIV-gp120 protein (arrows denote triple-positive cells), and (**E**) corresponds to the merging of all colors. (**F**) Quantified Macrophage/Microglia and astrocyte cells with integrated HIV DNA, producing HIV-mRNA and HIV-gp120 according to the viral replication status, including undetectable, low, and high replication (* *p* ≤ 0.005 compared to uninfected conditions, # *p* ≤ 0.005 compared to HIV_un_ conditions, and & *p* ≤ 0.005 compared to HIV_un_ or HIV_low_ conditions; n = 34 tissues analyzed and 21 tissues compared to uninfected tissues, Un and Alz, n = 8 different tissues; each point represents 3–5 different areas per tissue analyzed). HIVE total numbers were 1364 ± 279 cells, 876 ± 458 for macrophages, and 624 ± 287 for astrocytes under HIVE conditions (data not plotted). (**G**) Quantification of cells positive for HIV-gp120 detected in uninfected cells (HIV DNA (−)) surrounding the clusters containing integrated HIV DNA (HIV DNA (+)) corresponded to 750.61 ± 164.85 cells, suggesting bystander damage within the CNS (* *p* ≤ 0.005 compared to uninfected conditions and # *p* ≤ 0.005 compared to HIV DNA (+) cells). In contrast, only 10.45 ± 3.65 cells with integrated HIV DNA accumulated HIV-gp120 protein. In HIVE conditions, cells with integrated HIV DNA (HIVE-DNA (+)) were 1364 ± 349 cells, and uninfected cells (HIVE-DNA (−)) were 901 ± 302 cells; thus, local numbers of cells containing HIV-gp120 could be comparable to HIVE conditions but with a significantly lower expression or accumulation as well as being extremely localized. Bar: 100 µm.

**Figure 4 cells-11-02379-f004:**
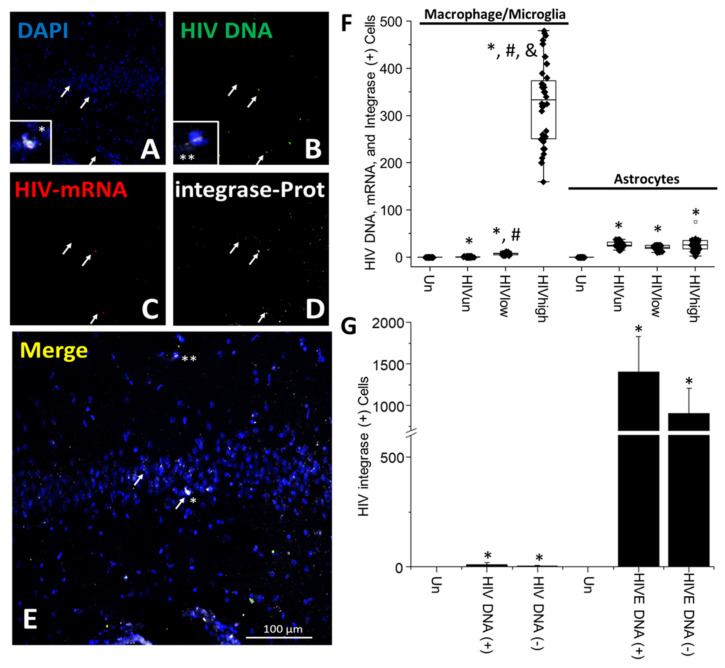
**HIV-integrase protein is poorly expressed in HIV-infected cells under ART.** (**A**–**E**) Representative confocal images showing the expression and distribution of integrase in the HIV-infected cells, showing minimal to no expression of integrase, where labeling corresponds to (**A**) DAPI, (**B**) HIV DNA, (**C**) HIV-mRNA, (**D**) HIV-integrase protein, and (**E**) merging of all colors. White arrows indicate cells with integrated HIV DNA. Asterisks and arrows indicate cells with high levels of cytoplasmic and nuclear integrase proteins. The insets indicate the colocalization of DAPI, HIV DNA, and integrase. (**F**) Quantification of cells with integrated HIV DNA, viral mRNA, and HIV-integrase protein according to its localization in Macrophage/Microglia or astrocytes in human brain tissues obtained from individuals with undetectable, low, and high viral replication (* *p* ≤ 0.005 compared to uninfected conditions, # *p* ≤ 0.005 compared to HIV_un_ conditions, and & *p* ≤ 0.005 compared to HIV_un_ or HIV_low_ conditions; n = 34 tissues analyzed and 21 tissues compared to uninfected tissues, Un and Alz, n = 8 different tissues; each point represents 3–5 different areas per tissue analyzed). The total numbers of cells in HIVE conditions were 1406 ± 425 cells, 905 ± 357 cells were macrophages, and 569 ± 299 cells were astrocytes (data not plotted). (**G**) Quantification of cells expressing HIV-integrase protein detected in HIV-positive and -negative cells, 11.22 ± 8.52 and 4.21 ± 2.46 cells, respectively. In contrast, in HIVE conditions, most integrase was accumulated in HIV DNA-positive cells, 1406 ± 425 cells (HIV DNA (+)), and HIV DNA-negative cells (HIV DNA (−)) 904 ± 305 cells, surrounding HIV-infected clusters. * *p* ≤ 0.005 compared to uninfected conditions. Bar: 100 µm.

**Figure 5 cells-11-02379-f005:**
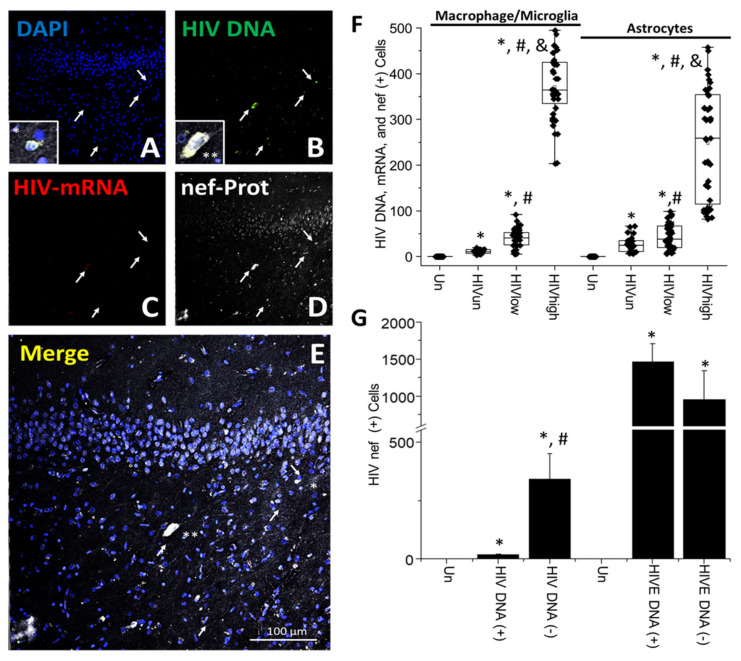
**HIV-nef is expressed in HIV-infected and surrounding uninfected cells, even in ART conditions.** (**A**–**E**) Representative confocal images show the expression and distribution of nef protein in the HIV-infected cells. Labeling corresponds to (**A**) DAPI, (**B**) HIV DNA, (**C**) HIV-mRNA, and (**D**) HIV-nef protein showing wide distribution in cells containing integrated HIV DNA (white arrows) and cells without integrated DNA, and (**E**) merging of all colors. The asterisks (* and **) and arrows represent cells with a high HIV-nef protein expression. The insets show cells with DAPI, HIV DNA, and HIV-nef protein. (**F**) Quantification of cells with integrated HIV DNA, viral mRNA, and HIV-nef protein in microglia/macrophages and astrocytes according to the tissue analyzed based on systemic replication, including undetectable, low, and high replication (* *p* ≤ 0.005 compared to uninfected conditions, # *p* ≤ 0.005 compared to HIV_un_ conditions, and & *p* ≤ 0.005 compared to HIV_un_ or HIV_low_ conditions; n = 34 tissues analyzed and 21 tissues compared to uninfected tissues, Un and Alz, n = 8 different tissues; each point represents 3–5 different areas per tissue analyzed). (**G**) Quantification of cells positive for HIV-nef protein in HIV-infected cells (HIV DNA (+)) was 17.15 ± 2.81 cells compared to neighboring uninfected cells (HIV DNA (−)) containing HIV-nef protein 341.63 ± 107.68 cells. This indicates a strong nef expression within the CNS in the current ART era, especially in neighboring uninfected cells. * *p* ≤ 0.005 compared to uninfected conditions, # *p* ≤ 0.005 compared to HIV_un_ conditions. Bar: 100 µm.

**Figure 6 cells-11-02379-f006:**
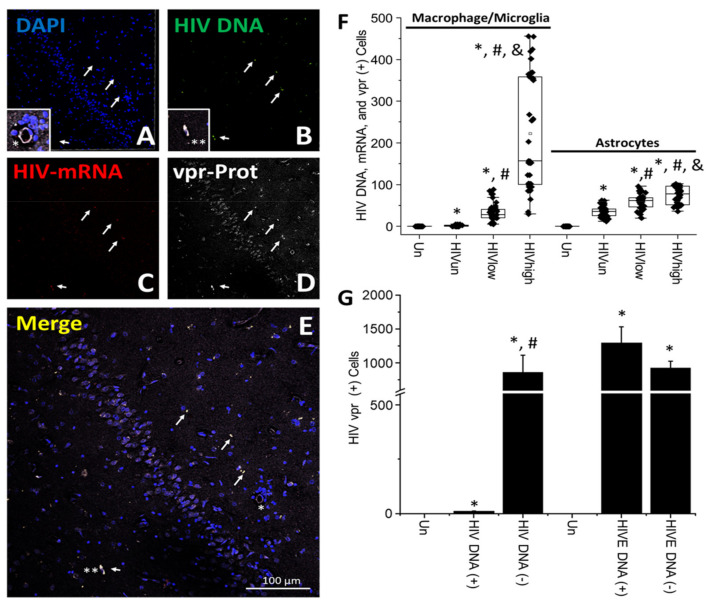
**HIV-vpr is expressed in HIV-infected cells under effective ART.** (**A**–**E**) Representative confocal images present the expression and distribution of Vpr protein in the HIV-infected cells. We identified that HIV-vpr was expressed in HIV-infected cells and cells without DNA. Labeling corresponds to (**A**) DAPI, (**B**) HIV DNA, (**C**) HIV-mRNA, and (**D**) HIV-vpr protein; it was not possible to establish a diffusion pattern similar to gp120 and nef proteins. HIV-vpr-positive cells were scattered within the brain (arrows represent HIV-infected cells producing HIV-vpr, * or ** represent cells with high HIV-vpr expression without an association to HIV-infected cells), (**E**) merging of all colors. The insets indicate cells positive for DAPI, HIV DNA, and vpr protein. (**F**) Quantification of cells with integrated HIV DNA, producing HIV-mRNA and HIV-vpr protein in Macrophage/Microglia and astrocytes according to the brain tissue analyzed based on systemic replication, including undetectable, low, and high replication (* *p* ≤ 0.005 compared to uninfected conditions, # *p* ≤ 0.005 compared to HIV_un_ conditions, and & *p* ≤ 0.005 compared to HIV_un_ or HIV_low_ conditions; n = 34 tissues analyzed and 21 tissues compared to uninfected tissues, Un and Alz, n = 8 different tissues; each point represents 3–5 different areas per tissue analyzed). In HIVE conditions, HIV-vpr colocalized mostly with HIV-infected cells with total numbers (HIVE) of 1289 ± 245, macrophages 920 ± 105 cells, and astrocyte 623 ± 277 positive cells. (**G**) Quantification of cells expressing HIV-vpr protein in uninfected cells (HIV DNA (−)) positive for vpr was 855.83 ± 257.45 cells around the clusters containing cells with integrated HIV DNA, compared to HIV DNA-positive cells (HIV DNA (+)) with 9.68 ± 2.70 cells, suggesting a bystander secretion and uptake (* *p* ≤ 0.005 compared to uninfected conditions and # *p* ≤ 0.005 compared to HIV DNA(+) cells). In HIVE conditions, quantification shows most vpr was in cells with integrated HIV DNA (HIVE-DNA (+)), with 1289 ± 245 cells, and minimal amounts present in uninfected cells (HIVE-DNA (−)), with 920 ± 105 cells. Bar: 100 µm.

**Figure 7 cells-11-02379-f007:**
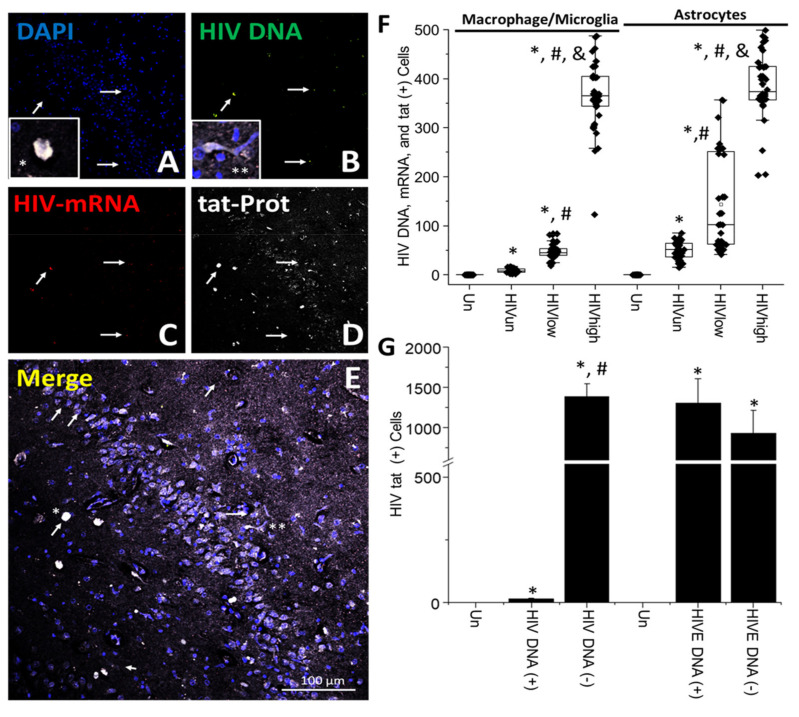
**HIV-tat protein is expressed within the CNS, even in the ART era.** (**A**–**E**) Representative confocal images show the expression and distribution of the tat protein in the HIV-infected cells and surrounding uninfected cells. Labeling corresponds to (**A**) DAPI, (**B**) HIV DNA, (**C**) HIV-mRNA, (**D**) HIV-tat protein showing diffusion from HIV DNA-positive cells into neighboring uninfected cells (white arrows indicate cells with integrated HIV DNA and */** indicates cells without integrated HIV DNA, but highly positive for HIV-tat), (**E**) merging of all colors. The insets indicate cells positive for DAPI, HIV DNA, and tat protein. (**F**) Quantification of cells with integrated HIV DNA, producing HIV-mRNA and HIV-tat in Macrophage/Microglia and astrocyte cells in brain tissues obtained from HIV-infected individuals with undetectable, low, and high replication (* *p* ≤ 0.005 compared to uninfected conditions, # *p* ≤ 0.005 compared to HIV_un_ conditions, and & *p* ≤ 0.005 compared to HIV_un_ or HIV_low_ conditions; n = 34 tissues analyzed and 21 tissues compared to uninfected tissues, Un and Alz, n = 8 different tissues; each point represents 3–5 different areas per tissue analyzed). In encephalitic conditions (HIVEs), HIV-tat colocalized mostly with HIV-infected cells, total numbers (HIVE, 1193 ± 244), Macrophage/Microglia (801 ± 197 cells), and a small population of astrocytes (685 ± 297 cells) (data not show). (**G**) Quantification of cells positive for tat protein-containing integrated HIV DNA (HIV DNA (+)) indicates only 13.96 ± 3.65 cells, whereas 1383.71 ± 164.85 uninfected surrounding cells (HIV DNA (−)) contained tat protein. A similar number of cells were found in HIV encephalitic brains (HIVEs), 1306 ± 306 cells expressed tat in HIV-infected cells (HIVE DNA (+)), and only 934 ± 277 cells without HIV DNA showed HIV-tat staining, HIV DNA (−), * *p* ≤ 0.005 compared to uninfected conditions, # *p* ≤ 0.005 compared to HIV DNA (+) cells. Bar: 100 µm.

**Table 1 cells-11-02379-t001:** **Patient information for the samples analyzed.** Information from all 34 individual donors of brain tissue samples regarding HIV status, age, sex, cognitive status, CD4^+^ T-cell counts (cells/mm^3^), CD8^+^ T cells (cells/mm^3^), viral load (copies/mL), years living with HIV, and ART treatment description. Patient Group—Un: uninfected/HIV-negative group; HIV_un_: HIV positive, viral load under the limit of detection; HIV_low_: HIV positive, low-viral-load group; HIV_high_: HIV positive, high-viral-load group; HIVE: HIV positive, archival encephalitic group; Alz: Alzheimer’s group. Cognitive status—MCMD: minor neurocognitive and motor disorder, HAD: HIV-associated dementia. ART: antiretroviral. CD4, CD8 T-cell counts—N.R.: not reported. Brain Section—C: cortex, H: hippocampus, B: cortex and hippocampus.

N^o^	Patient Group	HIV	Age	Sex	Cognitive Status	CD4 Cells/mm^3^	CD8, Cells/mm^3^	Viral Load, Copies/mL	Years with HIV	ART	Brain Area
1	Un	-	44	M	None	-	-	-	-	-	B
2	Un	-	50	M	None	-	-	-	-	-	B
3	Un	-	69	M	None	-	-	-	-	-	B
4	Un	-	38	M	None	-	-	-	-	-	C
5	Un	-	51	F	None	-	-	-	-	-	B
6	HIV_un_	+	70	M	MCMD	741	N.R.	<50	26	ziagen, epivir, fosamprenavir, ritonavir	B
7	HIV_un_	+	69	F	MCMD	616	1190	<50	22	ziagen, epivir, fosamprenavir	C
8	HIV_un_	+	61	F	MCMD	488	465	<50	24	efavirenz, ziagen, epivir	B
9	HIV_un_	+	68	F	None	1306	380	0	15	emtricitabine, tenofovir alafenamide, elvitegravir, cobicistat	B
10	HIV_un_	+	48	M	MCMD	1658	50	<50	13	etravirine, ritonavir, maraviroc, darunavir	C
11	HIV_un_	+	36	M	None	293	881	<50	11	zidovudine, lamivudine, nelfinavir	C
12	HIV_un_	+	59	M	None	428	768	<50	26	lamivudine, darunavir, raltegravir, ritonavir, etravirine	B
13	HIV_un_	+	43	F	None	282	376	<50	24	emtricitabine, tenofovir alafenamide, darunavir, cobicistat	C
14	HIV_low_	+	38	M	None	973	864	<400	14	etravirine, abacavir, zidovudine, lamivudine	B
15	HIV_low_	+	53	M	Prob. HAD	101	396	<400	18	lamivudine, nevirapine, zidovudine	C
16	HIV_low_	+	62	F	None	750	647	275	25	darunavir, ritonavir, emtricitabine, tenofovir	C
17	HIV_low_	+	51	M	Prob. MCMD	126	N.R.	349	1	lamivudine, abacavir, nelfinavir	B
18	HIV_low_	+	46	M	Prob. MCMD	214	1330	12,352	25	darunavir, raltegravir, etravirine, emtricitabine, tenofovir	B
19	HIV_low_	+	61	M	Prob. HAD	500	1077	<400	18	atazanavir, emtricitabine, nevirapine, ritonavir	C
20	HIV_high_	+	49	M	Prob. MCMD	3	71	>750,000	21	amprenavir, stavudine, lopinavir	C
21	HIV_high_	+	44	F	Prob. HAD	1	139	>750,000	9	stavudine, didanosine, nevirapine, nelfinavir	C
22	HIV_high_	+	42	M	Prob. HAD	37	1683	691,338	6	didanosine, efavirenz, nelfinavir	B
23	HIV_high_	+	37	F	Prob. HAD	1	1101	>750,000	3	zidovudine, didanosine, lopinavir, ritonavir	C
24	HIV_high_	+	40	F	Prob. HAD	14	N.R.	>750,000	4	indinavir, zidovudine	C
25	HIV_high_	+	34	M	Prob. HAD	10	545	165,862	6	zidovudine, nelfinavir, indinavir, didanosine, lamivudine, abacavir, saquinavir-sgc	B
26	HIV_high_	+	37	M	Prob. MCMD	91	N.R.	>750,000	2	efavirenz, lopinavir, ritonavir, nevirapine, tenofovir, abacavir, zidovudine, lamivudine	C
27	HIVE	+	43	M	HAD	7	N.R.	119,000	6	None	B
28	HIVE	+	40	F	HAD	5	N.R.	750,000	8	atazanavir	H
29	HIVE	+	43	M	HAD	21	N.R.	172,000	5	None	B
30	HIVE	+	38	M	HAD	185	N.R.	46,284	3	None	C
31	HIVE	+	44	F	HAD	64	N.R.	750,000	5	emtricitabine, tenofovir	B
32	Alz	-	62	F	Alz	-	-	-	-	-	C
33	Alz	-	68	M	Alz	-	-	-	-	-	C
34	Alz	-	62	M	Alz	-	-	-	-	-	C

**Table 2 cells-11-02379-t002:** **Detection of HIV DNA, HIV-mRNA, and HIV protein in uninfected HL-60 cells.** We diluted uninfected control cells (HL-60) to calculate our image system’s sensitivity to detect viral reservoirs. One to ten cells were diluted in 10^3^ to 10^12^ HeLa cells. Then, staining for HIV DNA (top), HIV-mRNA (middle), or HIV-p24 protein (bottom) was performed to identify whether we could detect the diluted cells in the HeLa cells. The percentage of detection is indicated for 10 sample preparations for each dilution. Pellets of these cells were cut and analyzed by microscopy to identify the infected cells.

		% Positive for HIV DNA									
		1	2	3	4	5	6	7	8	9	10
**Dilution (number of HeLa cells)**	**10^3^**	0	0	0	0	0	0	2	0	0	0
	**10^4^**	0	1	1	0	0	0	1	0	0	0
	**10^5^**	0	1	10	0	0	0	0	0	0	0
	**10^6^**	0	0	0	0	0	0	0	0	0	0
	**10^7^**	0	0	0	0	0	0	0	0	0	0
	**10^8^**	0	0	0	0	0	0	1	0	0	0
	**10^9^**	0	0	0	0	0	0	0	0	0	0
	**10^10^**	0	0	0	0	0	0	5	0	0	0
	**10^11^**	0	0	0	0	0	0	0	0	0	0
	**10^12^**	0	0	0	0	0	0	0	0	0	0
		**% Positive for HIV DNA + HIV-mRNA**									
		**1**	**2**	**3**	**4**	**5**	**6**	**7**	**8**	**9**	**10**
**Dilution (number of HeLa cells)**	**10^3^**	0	1	0	0	0	0	0	0	0	0
	**10^4^**	0	0	0	0	0	0	0	0	0	0
	**10^5^**	2	1	1	0	0	0	0	0	0	0
	**10^6^**	0	0	0	0	0	0	0	0	0	0
	**10^7^**	0	0	0	0	0	0	0	0	0	0
	**10^8^**	0	1	0	0	0	0	0	0	0	0
	**10^9^**	0	1	0	0	0	0	0	0	0	0
	**10^10^**	0	1	0	0	1	0	0	0	0	0
	**10^11^**	0	0	0	0	0	0	0	0	0	0
	**10^12^**	0	0	0	0	0	0	0	0	0	0
		**% Positive for HIV DNA + HIV-mRNA + p24**									
		**1**	**2**	**3**	**4**	**5**	**6**	**7**	**8**	**9**	**10**
**Dilution (number of HeLa cells)**	**10^3^**	0	0	0	0	0	0	2	0	0	0
	**10^4^**	0	1	0	0	0	0	6	3	0	0
	**10^5^**	0	1	1	1	1	0	0	0	0	0
	**10^6^**	0	0	0	0	0	0	0	0	0	0
	**10^7^**	0	0	0	0	0	0	0	0	0	0
	**10^8^**	0	0	0	0	0	0	0	0	0	0
	**10^9^**	0	1	0	0	0	0	0	0	0	0
	**10^10^**	0	0	0	0	0	0	0	0	0	0
	**10^11^**	0	0	0	0	0	0	0	0	0	0
	**10^12^**	0	0	0	0	0	0	0	0	0	0

**Table 3 cells-11-02379-t003:** **Detection of HIV DNA, HIV-mRNA, and HIV protein in uninfected A3.01 cells.** We diluted uninfected control cells (A3.01) to calculate our image system’s sensitivity to detect viral reservoirs. One to ten cells were diluted in 10^3^ to 10^12^ HeLa cells. Then, staining for HIV DNA (top), HIV-mRNA (middle), or HIV-p24 (bottom) was performed to identify whether we could detect the diluted cells in the HeLa cells. The percentage of detection is indicated for 10 sample preparations for each dilution. Pellets of these cells were cut and analyzed by microscopy to identify the infected cells.

		% Positive for HIV DNA									
		1	2	3	4	5	6	7	8	9	10
**Dilution (number of HeLa cells)**	**10^3^**	0	10	0	0	0	0	0	0	0	0
	**10^4^**	0	0	0	0	0	0	0	0	0	0
	**10^5^**	2	1	0	0	0	0	0	0	0	0
	**10^6^**	0	0	0	0	0	0	0	0	0	0
	**10^7^**	0	0	0	0	0	0	0	0	0	0
	**10^8^**	0	0	0	0	0	0	0	0	0	0
	**10^9^**	0	0	0	0	0	0	1	0	0	0
	**10^10^**	0	1	0	0	0	0	0	0	0	0
	**10^11^**	0	0	0	0	0	0	0	0	0	0
	**10^12^**	0	0	0	0	0	0	0	0	0	0
		**% Positive for HIV DNA + HIV-mRNA**									
**Dilution (number of HeLa cells)**		**1**	**2**	**3**	**4**	**5**	**6**	**7**	**8**	**9**	**10**
	**10^3^**	0	1	0	0	0	0	0	0	0	0
	**10^4^**	0	0	0	0	0	0	0	0	0	0
	**10^5^**	3	2	0	1	0	0	0	1	0	0
	**10^6^**	0	0	0	0	0	0	0	0	0	0
	**10^7^**	0	0	0	0	0	1	0	0	0	0
	**10^8^**	0	0	0	0	0	0	0	0	0	0
	**10^9^**	0	0	0	0	0	0	0	1	0	0
	**10^10^**	0	0	0	0	0	0	0	0	0	0
	**10^11^**	0	0	0	0	0	1	0	0	0	0
	**10^12^**	0	0	0	0	0	0	0	0	0	0
		**% Positive for HIV DNA + HIV-mRNA + p24**									
		**1**	**2**	**3**	**4**	**5**	**6**	**7**	**8**	**9**	**10**
**Dilution (number of HeLa cells)**	**10^3^**	0	0	0	0	0	0	0	0	0	0
	**10^4^**	0	0	0	0	0	0	0	0	0	0
	**10^5^**	1	1	0	0	0	0	0	0	0	0
	**10^6^**	0	0	0	0	0	0	0	0	0	0
	**10^7^**	0	0	0	0	0	0	0	0	0	0
	**10^8^**	0	0	0	0	0	0	0	0	0	0
	**10^9^**	0	0	0	0	0	0	0	0	0	0
	**10^10^**	0	0	0	0	0	0	0	0	0	0
	**10^11^**	0	0	0	0	0	0	0	0	0	0
	**10^12^**	0	0	0	0	0	0	0	0	0	0

**Table 4 cells-11-02379-t004:** **Detection of HIV DNA, HIV-mRNA, and HIV-protein in OM-10 cells.** To calculate our image system’s sensitivity to detect viral reservoirs, we diluted cells with one copy of integrated HIV DNA (OM-10). One to ten cells were diluted in 10^3^ to 10^12^ HeLa cells. Then, staining for HIV DNA (top), HIV-mRNA (middle), or HIV-p24 (bottom) was performed to identify whether we could detect the diluted cells in the HeLa cells. The percentage of detection is indicated for 10 sample preparations for each dilution. Pellets of these cells were cut and analyzed by microscopy to identify the infected cells. These cells were not treated with reactivating agents, such as TNF-α or SAHA. Thus, detection corresponds to the baseline viral production.

		% Positive for HIV DNA									
		1	2	3	4	5	6	7	8	9	10
**Dilution (number of HeLa cells)**	**10^3^**	88	80	78	100	83	100	100	100	100	100
	**10^4^**	89	94	93	100	83	100	100	98	100	95
	**10^5^**	78	94	100	85	83	97	95	95	100	98
	**10^6^**	83	94	100	100	77	97	98	100	100	100
	**10^7^**	100	93	97	100	77	97	93	98	100	100
	**10^8^**	100	85	100	90	70	95	83	98	100	100
	**10^9^**	100	100	100	100	77	100	95	93	100	100
	**10^10^**	100	92	100	90	80	100	100	100	100	100
	**10^11^**	57	100	100	90	83	100	100	100	100	98
	**10^12^**	80	89	100	95	80	97	100	100	100	100
		% **Positive for HIV DNA + HIV-mRNA**									
		**1**	**2**	**3**	**4**	**5**	**6**	**7**	**8**	**9**	**10**
**Dilution (number of HeLa cells)**	**10^3^**	100	70	84	100	83	100	100	100	100	100
	**10^4^**	100	100	89	100	80	99	99	98	100	95
	**10^5^**	86	100	100	85	83	97	95	95	100	98
	**10^6^**	100	100	100	100	75	96	98	100	100	99
	**10^7^**	100	100	100	100	77	97	93	96	100	100
	**10^8^**	100	100	100	90	69	95	83	96	100	95
	**10^9^**	100	100	100	100	77	100	95	93	100	99
	**10^10^**	100	100	100	90	99	100	100	100	100	100
	**10^11^**	80	100	100	90	83	100	100	100	100	98
	**10^12^**	100	100	100	95	79	97	100	100	100	100
		**% Positive for HIV DNA + HIV-mRNA + p24**									
		**1**	**2**	**3**	**4**	**5**	**6**	**7**	**8**	**9**	**10**
**Dilution (number of HeLa cells)**	**10^3^**	88	69	85	100	83	100	100	100	100	99
	**10^4^**	100	100	88	100	83	100	99	98	100	96
	**10^5^**	85	94	100	88	83	97	96	95	100	97
	**10^6^**	100	94	100	100	75	97	98	100	100	100
	**10^7^**	100	93	96	98	77	97	91	95	100	100
	**10^8^**	100	85	100	90	69	95	81	98	100	96
	**10^9^**	100	100	100	100	77	100	95	90	100	99
	**10^10^**	100	92	100	90	79	100	100	100	100	100
	**10^11^**	81	100	100	88	83	100	100	99	100	98
	**10^12^**	80	89	100	95	81	97	100	100	100	99

**Table 5 cells-11-02379-t005:** **Detection of HIV DNA, HIV-mRNA, and HIV protein in ACH-2 cells.** To calculate our image system’s sensitivity to detect viral reservoirs, we diluted cells with one copy of integrated HIV DNA (ACH-2), one to ten cells into 10^3^ to 10^12^ HeLa cells. Then, staining for HIV DNA (top), HIV-mRNA (middle), or HIV-p24 (bottom) was performed to identify whether we could detect the diluted cells in the HeLa cells. The percentage of detection is indicated for 10 sample preparations for each dilution. Pellets of these cells were cut and analyzed by microscopy to identify the infected cells. These cells were not treated with reactivating agents such as TNF-α or SAHA. Thus, detection corresponds to the baseline viral production.

		% Positive for HIV DNA									
		1	2	3	4	5	6	7	8	9	10
**Dilution (number of HeLa cells)**	**10^3^**	85	77	95	100	100	100	91	91	100	100
	**10^4^**	79	83	93	100	92	100	94	91	92	100
	**10^5^**	100	81	93	100	88	100	86	94	89	95
	**10^6^**	100	76	95	100	100	96	91	91	86	93
	**10^7^**	100	84	100	100	100	90	93	100	92	90
	**10^8^**	100	83	93	85	80	93	89	100	92	98
	**10^9^**	100	83	87	90	84	100	86	88	89	100
	**10^10^**	100	81	88	95	88	87	75	75	95	100
	**10^11^**	100	83	93	89	100	90	87	87	89	100
	**10^12^**	100	83	89	81	84	87	87	87	100	100
		**% Positive for HIV DNA + HIV-mRNA**									
**Dilution (number of HeLa cells)**		**1**	**2**	**3**	**4**	**5**	**6**	**7**	**8**	**9**	**10**
	**10^3^**	84	76	91	100	100	100	90	92	100	100
	**10^4^**	80	83	93	100	93	100	94	89	92	100
	**10^5^**	100	81	94	100	87	100	88	95	89	93
	**10^6^**	100	76	95	100	100	96	89	89	86	93
	**10^7^**	100	84	100	100	92	91	94	100	92	91
	**10^8^**	100	83	94	85	100	94	89	100	91	98
	**10^9^**	100	83	100	89	84	100	86	89	89	100
	**10^10^**	100	79	88	95	85	97	77	77	94	100
	**10^11^**	100	83	93	88	100	93	93	93	88	99
	**10^12^**	100	83	88	87	83	100	100	100	100	100
		**% Positive for HIV DNA + HIV-mRNA + p24**									
		**1**	**2**	**3**	**4**	**5**	**6**	**7**	**8**	**9**	**10**
**Dilution (number of HeLa cells)**	**10^3^**	100	76	90	100	100	100	91	90	100	100
	**10^4^**	81	83	93	100	91	100	94	90	92	100
	**10^5^**	86	80	91	100	88	100	86	89	90	92
	**10^6^**	100	73	96	100	100	94	90	90	91	91
	**10^7^**	100	83	100	100	90	92	93	100	92	88
	**10^8^**	100	82	92	98	100	93	89	100	90	89
	**10^9^**	100	83	100	100	82	100	86	88	90	100
	**10^10^**	100	80	89	100	88	97	75	75	95	100
	**10^11^**	83	83	94	100	100	93	87	94	90	99
	**10^12^**	67	82	88	100	84	100	100	100	100	100

## Data Availability

The data sets generated are all available upon reasonable request.

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
