# Peer review of "Identification, Quantification, and Characterization of HIV-1 Reservoirs in the Human Brain"

_cells, 2022, doi:10.3390/cells11152379_

Round 1
Reviewer 1 Report
Characterization and quantification of HIV-1 reservoirs in the CNS are crucial and timely topics, especially in the light of recent advances in gene-editing strategies for HIV cure. The manuscript's authors describe a novel imaging multi-component method, using large brain sections to identify and quantify in a cell-specific manner viral reservoir in the human brain. Using samples stratified according to ART status and viral loads in blood, the authors demonstrated that although long-lasting art reduced the abundance of HIV-1 infected cells in the brain, it did not prevent residual production of viral mRNA and proteins. This effect was primarily due reduction of the pool of infected myeloid cells, while the infected astrocytes were only minimally reduced by long-term ART. Interestingly, authors show accumulation of several viral proteins in uninfected cells surrounding clusters of HIV-1 infected cells suggesting protein secretion bystander uptake. Unfortunately, no correlation between the number of HIV-1 positive cell clusters and the degree of cognitive decline was observed, but this type of analysis requires a much larger number of subjects. Below are some major and minor comments to the authors.
1) Clarify how ratios and % of cell clusters positivity were calculated.
Line 526, approximately 1/3 of cell clusters expressing HIV-DNA could express viral mRNA but based on data shown in Fig 1, this is true for HIVun (0.86/2.67=0.32) and HIVlow (1.88/5.38=0.35) groups, but for HIVhigh group it is over 2/3 (14.22/22.4=0.63). Similarly, Line 534 states that 1/3 cells containing viral DNA and RNA were positive for HIV-1 p24 but for HIVud 0.37/0.86=0.43 (43%), HIVlow 1.16/1.88=0.62 (62%), and HIV high 9.88/14.22=0.69 (69%). Line 542 states that the number of clusters of HIV-infected tissues decreased about 2/3 compared to HIVhigh and HIVE conditions; based on what calculations? For example, using HIV DNA positive clusters that decrease will be higher: HIVlow/HIVhigh (5.38/22.4=0.241, 75.9% drop) HIVud/high (2.67/22.4=0.12, 82% drop).
2) The discrepancy between the number of triple positive cells for different viral protein stainings/tissue sections?.
Figure 2 shows the quantification of myeloid and glial viral reservoirs. HIV DNA, HIV DNA + mRNA, triple positive HIV DNA, mRNA, and p24 positive cells are shown. They follow the trend: DNA positive>DNA+mRNA positive>DNA+mRNA+p24 positive. For other viral proteins Figure 3-7, only triple-positive cell numbers are shown, but many of them for myeloid cells and astrocytes from the HIVhigh group are higher than the number of HIV DNA-positive cells shown in Fig 2. For example: there are 307 HIV DNA positive macrophage/microglia in HIVhigh group (Fig 2 I) but 325 (for DNA/mRNA/integrase, Fig 4F), 371 (for DNA/mRNA/nef, Fig 5F) and 366 (for DNA/mRNA/tat, Fig 7F). Similarly, there are 44 DNA-positive astrocytes in the HIVhigh group (Fig 2 I) and 247 (for DNA/mRNA/nef, Fig 5F), and 385 (for DNA/mRNA/tat, Fig 7F). That indicates that the number of HIV DNA and HIV DNA + mRNA positive cells detected varied between serial tissue sections. It is stated the HIV-DNA nef, Alu repeats, and DAPI stainings were repeated in all 9 sections to reposition brain structures, clusters of HIV-infected cells, and cell types identified (as shown in Figure 1). That would indicate that the numbers of HIV DNA and HIV DNA+mRNA positive cells detected varied between serial sections, and the data from Figure 2 underestimate the total numbers of HIV DNA and HIV DNA+mRNA positive cells detected. Can you comment on this? Can you provide a summary Table showing all the mean values for the number of cells positive for HIV DNA, HIV DNA + mRNA, HIV DNA + mRNA + viral protein from each staining if available (p24, gp120, Integrase, nef, vpr, tat)?
The presence of p24 in neighboring cells is intriguing since most of the p24 capsid protein (>90%) is associated with HIV-1 particles. So, can Gag p24 be secreted?
The observed poor expression of Integrase is likely due to a much lower translation rate (ribosomal frameshifting producing Gag-Pol precursor protein occurs at 10-20 times lower frequency than Gag precursor p55 protein translation).
Why were all groups (HIVud, HIVlow, HIVhigh) combined for calculating bystander effects? The data suggest significant differences in the expression of all viral proteins between HIVud/HIVlow and HIVhigh groups. Were the numbers of HIV DNA (-) viral protein (+) cells similar between these groups?
Did bystander uptake correlate with cognitive decline?
Can the observed in astrocytes differences in response to long-term ART be related to viral RNA splicing? For example, gag p24 and Integrase are expressed from unspliced viral mRNA (unchanged) while Vpr from single spliced and Tat and Nef from multiply spliced viral RNAs (ART responsive).
Line 419 states that there was no significant difference in sex between HIV-negative and HIV-positive groups, but the numbers suggest otherwise (HIV-1-positive: female=38%, male=62%, HIV-1-negative: female=20%, male 80%)
Table 2-5. It is not clear how these dilutions and data analyses were made. The percentage detection is indicated for 10 wells for each dilution. What kind of wells can hold 10^12 HeLa cells? How can you detect 1 or 10 HIV-positive cells in 10^12 negative ones? What was the limit of detection for each assay? Are these tables necessary?
Additionally, supplementary Figure 12 shows that only a small fraction of the latently infected cell lines OM-10 and ACH-2 expressed viral RNA and p24 (6.5% and 9.2% of triple positive, respectively). Tables 4 and 5 show close to 100% detection of viral RNA and p24 positive OM-10 and ACH2 cells diluted with HeLa cells. Were cells pre-treated with PMA or TNFa used for these dilutions? If yes, it should be indicated in the figure legends.
Table 4 and 5 heading: change "in uninfected OM-10/ACH-2" to infected.
Correct statements below:
Line 314 "experiments using the recombinant experiments"
Line 316 "After these controls were properly passed all our quality controls"
Author Response
Reviewer #1: Characterization and quantification of HIV-1 reservoirs in the CNS are crucial and timely topics, especially in the light of recent advances in gene-editing strategies for HIV cure. The manuscript's authors describe a novel imaging multi-component method, using large brain sections to identify and quantify in a cell-specific manner viral reservoir in the human brain. Using samples stratified according to ART status and viral loads in blood, the authors demonstrated that although long-lasting art reduced the abundance of HIV-1 infected cells in the brain, it did not prevent residual production of viral mRNA and proteins. This effect was primarily due reduction of the pool of infected myeloid cells, while the infected astrocytes were only minimally reduced by long-term ART. Interestingly, authors show accumulation of several viral proteins in uninfected cells surrounding clusters of HIV-1 infected cells suggesting protein secretion bystander uptake. Unfortunately, no correlation between the number of HIV-1 positive cell clusters and the degree of cognitive decline was observed, but this type of analysis requires a much larger number of subjects. Below are some major and minor comments to the authors.
Answer: Thank you, and the reviewer is totally correct; we are collecting more samples to correlate cognitive impairment and the size of the viral reservoirs as well as with long-term ART.
Reviewer #1: 1) Clarify how ratios and % of cell clusters positivity were calculated. Line 526, approximately 1/3 of cell clusters expressing HIV-DNA could express viral mRNA but based on data shown in Fig 1, this is true for HIVun (0.86/2.67=0.32) and HIVlow (1.88/5.38=0.35) groups, but for HIVhigh group it is over 2/3 (14.22/22.4=0.63). Similarly, Line 534 states that 1/3 cells containing viral DNA and RNA were positive for HIV-1 p24 but for HIVun 0.37/0.86=0.43 (43%), HIVlow 1.16/1.88=0.62 (62%), and HIV high 9.88/14.22=0.69 (69%). Line 542 states that the number of clusters of HIV-infected tissues decreased about 2/3 compared to HIVhigh and HIVE conditions; based on what calculations? For example, using HIV DNA positive clusters that decrease will be higher: HIVlow/HIVhigh (5.38/22.4=0.241, 75.9% drop) HIVun/high (2.67/22.4=0.12, 82% drop).
Answer: This is an excellent point, that we clarify in the text. Thank you, great comment. The 2/3 did not compare different conditions only compared the same condition, HIV DNA, HIV-mRNA, and respective viral protein in undetectable, low, and high. If we compare the data in the way, as you described, the drop is highly significant. However, when we presented these data at conferences, most questions were related to how significant the areas were analyzed (cortex and hippocampus) compared to other areas that may not be affected by ART. Thus, we decided to be careful in our data presentation and assessments because we also considered these possibilities. However, we included the analysis and the disclosure of these points. Thank you!
Reviewer #1: 2) The discrepancy between the number of triple positive cells for different viral protein staining/tissue sections?. Figure 2 shows the quantification of myeloid and glial viral reservoirs. HIV DNA, HIV DNA + mRNA, triple positive HIV DNA, mRNA, and p24 positive cells are shown. They follow the trend: DNA positive>DNA+mRNA positive>DNA+mRNA+p24 positive. For other viral proteins Figure 3-7, only triple-positive cell numbers are shown, but many of them for myeloid cells and astrocytes from the HIVhigh group are higher than the number of HIV DNA-positive cells shown in Fig 2. For example: there are 307 HIV DNA positive macrophage/microglia in HIVhigh group (Fig 2 I) but 325 (for DNA/mRNA/integrase, Fig 4F), 371 (for DNA/mRNA/nef, Fig 5F) and 366 (for DNA/mRNA/tat, Fig 7F). Similarly, there are 44 DNA-positive astrocytes in the HIVhigh group (Fig 2 I) and 247 (for DNA/mRNA/nef, Fig 5F), and 385 (for DNA/mRNA/tat, Fig 7F). That indicates that the number of HIV DNA and HIV DNA + mRNA positive cells detected varied between serial tissue sections. It is stated the HIV-DNA nef, Alu repeats, and DAPI stainings were repeated in all 9 sections to reposition brain structures, clusters of HIV-infected cells, and cell types identified (as shown in Figure 1). That would indicate that the numbers of HIV DNA and HIV DNA+mRNA positive cells detected varied between serial sections, and the data from Figure 2 underestimate the total numbers of HIV DNA and HIV DNA+mRNA positive cells detected. Can you comment on this? Can you provide a summary Table showing all the mean values for the number of cells positive for HIV DNA, HIV DNA + mRNA, HIV DNA + mRNA + viral protein from each staining if available (p24, gp120, Integrase, nef, vpr, tat)?
Answer: You are totally right and thank you so much for the great comment. The Alu-repeats and HIV-nef were done every 7-8 sections. Everything in between these sections has HIV DNA in the current manuscript. Our future manuscript considers this essential point, especially in individuals with long-term ART, where the viral reservoir size decrease and could be underestimated. However, the observation is more complicated because some cells accumulate specific proteins, including HIV-tat and HIV-nef, that mostly accumulate in neurons. We include this great point in the discussion. Thank you!
Reviewer #1: The presence of p24 in neighboring cells is intriguing since most of the p24 capsid protein (>90%) is associated with HIV-1 particles. So, can Gag p24 be secreted?
Answer: We agreed with you. Due to this issue, we pursued this point in astrocytes for a long time. In astrocytes, we extensive looks to correlate viral presence (even immature virions) with the virus, and we identified a highly localized secretion of virus that was undetectable for P24 or COBAS test, with poor correlation. We start working with Dr. Ono (University of Michigan) to understand this issue. But still is an ongoing question. Thank you so much. Great comment.
Reviewer #1: The observed poor expression of Integrase is likely due to a much lower translation rate (ribosomal frameshifting producing Gag-Pol precursor protein occurs at 10-20 times lower frequency than Gag precursor p55 protein translation).
Answer: We agreed with the reviewer. We include this point in the discussion.
Reviewer #1: Why were all groups (HIVun, HIVlow, HIVhigh) combined for calculating bystander effects? The data suggest significant differences in the expression of all viral proteins between HIVun/HIVlow and HIVhigh groups. Were the numbers of HIV DNA (-) viral protein (+) cells similar between these groups?
Answer: We did not expand this point for a critical issue not discussed in the manuscript due to the differences in bystander uptake that needs significant calculations, not included in the current manuscript, including a differential uptake by neurons and macrophages for HIV-tat and HIV-nef in astrocytes and less in neurons. Thus, now we are expanding the quantification and characterization of the uptake into the surrounding cells, including neurons, astrocytes, EC, pericytes, and macrophage/microglia. However, this data is still under analysis due to the significant variability in secretion, uptake, and cognitive impairment. Further to complicate the issue, drug use also contributes to variability preventing the reduction in the viral reservoir pool and enhancing viral protein secretion. These issues will be part of future manuscripts. Thank you, a great comment.
Reviewer #1: Did bystander uptake correlates with cognitive decline?
Answer: Yes and no; let me explain why we did not include this point. Yes, if we analyzed the brains of HIV-infected individuals with worse drug use and cognitive impairment, there was a correlation with the size of the viral reservoir (clusters and cells per cluster). Further, we identified that drug use prevented the decrease in the viral reservoir pool in the brain and lymph nodes. However, why not too.
Reviewer #1: Can the observed in astrocytes differences in response to long-term ART be related to viral RNA splicing? For example, gag p24 and Integrase are expressed from unspliced viral mRNA (unchanged) while Vpr from single spliced and Tat and Nef from multiply spliced viral RNAs (ART responsive).
Answer: Excellent point; we do not know. This point will be great to examine in future experiments. Thank you for the suggestion. We will include these approaches in the new quantifications. Thank you!
Reviewer #1: Line 419 states that there was no significant difference in sex between HIV-negative and HIV-positive groups, but the numbers suggest otherwise (HIV-1-positive: female=38%, male=62%, HIV-1-negative: female=20%, male 80%)
Answer: Thank you, we miss that point.
Reviewer #1: Table 2-5. It is not clear how these dilutions and data analyses were made. The percentage detection is indicated for 10 wells for each dilution. What kind of wells can hold 10^12 HeLa cells? How can you detect 1 or 10 HIV-positive cells in 10^12 negative ones? What was the limit of detection for each assay? Are these tables necessary?
Answer: We clarify this point in the current version. We cultured the cells and added them to a large tube, generating a small or large pellet to cut, top to bottom, to identify cells with HIV products. We expanded this section.
Reviewer #1: Additionally, supplementary Figure 12 shows that only a small fraction of the latently infected cell lines OM-10 and ACH-2 expressed viral RNA and p24 (6.5% and 9.2% of triple positive, respectively). Tables 4 and 5 show close to 100% detection of viral RNA and p24 positive OM-10 and ACH2 cells diluted with HeLa cells. Were cells pre-treated with PMA or TNFa used for these dilutions? If yes, it should be indicated in the figure legends.
Answer: We revised the figure legend.
Reviewer #1: Table 4 and 5: change "in uninfected OM-10/ACH-2" to infected.
Answer: Thank you, we miss that.
Reviewer #1: Correct statements below:
Line 314 "experiments using the recombinant experiments"
Line 316 "After these controls were properly passed, all our quality controls"
Answer: Thank you, we corrected the issues.
Reviewer 2 Report
The presence HIV reservoir in various tissues in ART era represents as a major obstacle to cure HIV. Brain has been implicated as HIV reservoir, which contributes to the persistence of high prevalence of HIV-associated neurocognitive disorders. However, the size, identity, and localization of these HIV reservoirs in brain are not clear. In this elegantly designed study, Donoso et al. developed a novel reliable method capable of detecting HIV reservoir in post-mortem human brain samples with accuracy. Using this method, the authors quantified the cell type and frequency of HIV-infected cells in brain tissues and demonstrated convincingly that macrophages/microglia and astrocytes are HIV-1 reservoirs that still produce residual viral mRNA and proteins despite ART.
Although the manuscript is elegantly designed and well written, and findings are significant to the field, I have concerns regarding the detection of HIV-1 proteins.
It appears that gp120 (Figure 3E) Nef (Figure 5E), Vpr (Figure 6E), and Tat (Figure 7E) are expressed in almost every DAPI positive cells. Can you present us the expression of these proteins on both microglia (Iba-1) and astrocytes (GFAP), as representative images (at least in one of the HIV categories) of quantified data shown in Figure 3F, 5F, 6F, 7F. I can only find one representative image showing the co-expression of gp120 with Iba-1 in Supplemental Figure 3.
The representative images and quantified data (Figure 3, 5, 6, and 7) also indicate that these viral proteins are not just expressed in microglia or astrocytes, and they might be expressed on other CNS cells (possibly on hippocampal pyramidal neurons as indicated in Figure 5E, 6E, and 7E). Can you comment on such a possibility and potential consequence?
Such abundant expression of HIV proteins in brain (as shown in the representative images in Figure 3, 5, 6, and 7) seems contradict with literature indicating that it is not easy to locate these viral proteins in immuno-stained human brain samples.
Please specify the catalog numbers for the following antibodies obtained from NIH repository that have demonstrated specificity: α-HIV nef, α-HIV tat, α-HIV integrase, α-HIV gp120, α-HIV vpr?
Line 98, please double check the wording ‘HIV-integrated DNA’.
Author Response
We want to thanks both reviewers for the constructive comments and the time and detail to read the long manuscript. We believe, as both reviewers indicated, that this manuscript is essential to understand the chronic CNS damage observed in the HIV infected population. Thank you
Reviewer #2: The presence HIV reservoir in various tissues in ART era represents as a major obstacle to cure HIV. Brain has been implicated as HIV reservoir, which contributes to the persistence of high prevalence of HIV-associated neurocognitive disorders. However, the size, identity, and localization of these HIV reservoirs in brain are not clear. In this elegantly designed study, Donoso et al. developed a novel reliable method capable of detecting HIV reservoir in post-mortem human brain samples with accuracy. Using this method, the authors quantified the cell type and frequency of HIV-infected cells in brain tissues and demonstrated convincingly that macrophages/microglia and astrocytes are HIV-1 reservoirs that still produce residual viral mRNA and proteins despite ART.
Answer: Thank you
Reviewer #2: Although the manuscript is elegantly designed and well written, and findings are significant to the field, I have concerns regarding the detection of HIV-1 proteins.
It appears that gp120 (Figure 3E) Nef (Figure 5E), Vpr (Figure 6E), and Tat (Figure 7E) are expressed in almost every DAPI positive cells. Can you present us the expression of these proteins on both microglia (Iba-1) and astrocytes (GFAP), as representative images (at least in one of the HIV categories) of quantified data shown in Figure 3F, 5F, 6F, 7F. I can only find one representative image showing the co-expression of gp120 with Iba-1 in Supplemental Figure 3.
Answer: We agreed with the reviewer that expression looks a lot, but is not. We selected the areas positive for viral reservoirs, but most of the tissue is negative. We denote this point in the current version of the manuscript. In addition, for the last point, we are preparing a different manuscript to address the point of viral protein uptake in glial cells and neurons, because there is an striking difference in the amount of viral protein as well as intracellular distribution. We include a section in the discussion about this point as well as described the supplemental material in a better manner. Thank you
Reviewer #2: The representative images and quantified data (Figure 3, 5, 6, and 7) also indicate that these viral proteins are not just expressed in microglia or astrocytes, and they might be expressed on other CNS cells (possibly on hippocampal pyramidal neurons as indicated in Figure 5E, 6E, and 7E). Can you comment on such a possibility and potential consequence?
Answer: Thank you, we included this important issue in the results and discussion section
Reviewer#2: Such abundant expression of HIV proteins in brain (as shown in the representative images in Figure 3, 5, 6, and 7) seems contradict with literature indicating that it is not easy to locate these viral proteins in immuno-stained human brain samples.
Answer: We clarify the highly localized expression that do not correspond to the entire brain. Thank you
Reviewer #2: Please specify the catalog numbers for the following antibodies obtained from NIH repository that have demonstrated specificity: α-HIV nef, α-HIV tat, α-HIV integrase, α-HIV gp120, α-HIV vpr?
Answer: We added the current ID numbers for the antibodies.
Reviewer #2: Line 98, please double check the wording ‘HIV-integrated DNA’.
Answer: we fixed the point. Thank you
Reviewer 3 Report
This study describes important information and critical findings concerning HIV reservoirs in the human brain, detailing a method to detect HIV-DNA, viral mRNA, and proteins in a cell type-dependent manner and with high spatial resolution in the brain. The study has great publication merit, as it deals with difficult-to-obtain human samples, has a robust methodology, and brings essential findings to advance HIV research in an era of effective antiviral therapies and chronic infection. I have just two specific considerations that I would like the authors to take into account when reviewing the manuscript:
1) One of the current problems HIV-infected individuals face is chronic inflammation, even when they are on ART and have an undetectable viral load. One of the possible explanations for this problem is the residual replication, directly connected to the HIV reservoirs. I suggest that the authors mention this point in the introduction section and link the study implications on this issue in the discussion section (I believe that inflammation may gain more attention in the discussion beyond the points already raised by the authors).
2) The study approval number/code by the ethics committee should be detailed in the ethical aspects section (Institutional Review Board Statement).
Author Response
We want to thank both reviewers for the constructive comments and the time and detail to read the long manuscript. We believe, as both reviewers indicated, that this manuscript is essential to understand the chronic CNS damage observed in the HIV infected population.
Reviewer #3: This study describes important information and critical findings concerning HIV reservoirs in the human brain, detailing a method to detect HIV-DNA, viral mRNA, and proteins in a cell type-dependent manner and with high spatial resolution in the brain. The study has great publication merit, as it deals with difficult-to-obtain human samples, has a robust methodology, and brings essential findings to advance HIV research in an era of effective antiviral therapies and chronic infection. I have just two specific considerations that I would like the authors to take into account when reviewing the manuscript:
Answer: Thank you
Reviewer #3: 1) One of the current problems HIV-infected individuals face is chronic inflammation, even when they are on ART and have an undetectable viral load. One of the possible explanations for this problem is the residual replication, directly connected to the HIV reservoirs. I suggest that the authors mention this point in the introduction section and link the study implications on this issue in the discussion section (I believe that inflammation may gain more attention in the discussion beyond the points already raised by the authors).
Answer: We include this essential point in the introduction and expanded in the discussion.
Reviewer #3: 2) The study approval number/code by the ethics committee should be detailed in the ethical aspects section (Institutional Review Board Statement).
Answer: We included the information requested.
Round 2
Reviewer 2 Report
All my concerns have adequately addressed.